# Systems Dynamics and the Analytical Network Process for the Evaluation and Prioritization of Green Projects: Proposal That Involves Participative Integration

**Julian Andres Castrillon-Gomez** [1] , **Gerard Olivar-Tost** [2] **and Johnny Valencia-Calvo** [2,*]

1 Departamento de Ingeniería Eléctrica y Electrónica, Universidad Nacional Colombia Sede Manizales, Manizales 170001, Colombia
2 Departamento de Ciencias Naturales y Tecnología, Universidad de Aysén, Coyhaique 5950000, Chile
* Correspondence: johnny.valencia@uaysen.cl; Tel.: +56-957564225

**Abstract:** This research proposes a methodology for evaluating and prioritizing green projects based on an integrated approach between system dynamics modeling and the analytical network process. The methodology is presented in three stages: First of all, we show the citizen factors and obtain the data from the zone through community participation. Then, in the second stage, the model of system dynamics is consolidated and calibrated, which allows the generation of relevant information for experts by simulating model variables. In the third stage, considering the dependency and feedback relationships of the system, the model is translated to a complex network of many opinions which makes the decision making through peer review easier. The application of the methodology is presented using a case study undertaken in the California county that belongs to the Magdalena region in Colombia. The results allow to conclude that the proposed methodology makes the evaluation process and the prioritization of the projects easier because it is possible to advise the experts with respect to the variables that maximize investments and based on this select environmental initiatives that maximize investments and the environmental, social, or economic initiatives that best respond to the needs of the community. Likewise, we can demonstrate that the methodology can be applied to any rural community adjusting parameters and calibration variables to reflect the new conditions, both for natural resources and for particular policies and actions in such a way that the most appropriate project can be selected.

**Keywords:** community participation; system dynamics model; sustainable development; multi-criteria decision-making (MCDM); analytic network process (ANP)

## 1. Introduction

In Colombia, the era of the postwar period has opened the door for public and private resources to finance development projects in the affected communities. The set of agreements expressed in [1] are focused on contributing to the realization of human rights, so citizen participation in the construction of peace is very important, and likewise, as proposed in [2], we can make plans during the tasks of planning, execution, and monitoring of plans and programs in a way that guarantees socio-economic sustainability; likewise, the good use of natural resources such as water, rural ground in the general of axes of territorial development.

However, as we can see in [3], when we talk about sustainability and green projects, the importance is highlighted not only for protecting the areas of environmental interest but also for articulating efforts to improve the labor, economic, and welfare conditions for men and women belonging to these communities so we can contribute the reduction of the gap between urban and rural areas. Therefore, for the evaluation and prioritization of projects, it is necessary to start from the review and analysis of the conditions, needs, and characteristics of an environmental, economic, social, and cultural nature, which make each

territory and each community, as we said in [4], unique scenarios that must be considered so that decision-making is carried out in a comprehensive, coordinated, and adapted way for each case.

Therefore, the evaluation and prioritization of projects, in the postwar period in Colombia, are shown as the dilemma of where to direct investment decisions by not only local governments but also the central government, with the view of an effective reparation of the damage caused to the community [5], and, therefore, allows to overcome the backwardness of decades of war and inattention of the state [6].

However, often current rural and urban agendas respond more to political and/or economic commitments and obligations, and the problems, needs, and expectations of the community are left aside [7]. The above allows us to formulate the hypothesis of this research, according to which it is possible to evaluate and prioritize green projects in communities of the postwar period, using a multivariate participatory model (qualitative and quantitative) that integrates social knowledge and through which relevant information is generated at a social, environmental, and economic level, so it can be used by the experts to guide decision making.

In this way, the proposed methodology proposes, in the first instance, the construction of a model of system dynamics [8,9] that allows to integrate citizen factors (determined through the participatory rural diagnosis [10,11]) and generate information in the short, medium, and long term that can facilitate decision making and, in the second instance, the translation to a complex network of multiple criteria that allows the evaluation and prioritization of alternatives through peer review, as exposed by the analytical network process method (ANP) [12,13].

At present, there are limited applications where the modeling methodologies in system dynamics are integrated with the multicriteria evaluation ANP [14], and it has been very difficult to find research related to the evaluation and prioritization of projects.

The document is structured as follows: Section 2 shows the model of system dynamics; in Section 3 we can see the creation of the structure of the ANP network starting with the identification of the clusters and nodes in the model of system dynamics; Section 4 displays the application of the evaluation and prioritization methodology through a case study; afterwards, Section 5 shows the results of the methodology; then, Section 6 presents the discussion; and, finally, Section 7 presents the conclusion and comments regarding the perspective of the future research.

## 2. System Dynamic Model

The proposed methodology was carried out using a two-stage structure. Figure 1 shows the general approach of its application and adaptation to the specific case study. The first stage consists of the elaboration of the model using system dynamics through different steps.

The first step in the creation of the modeling is the problem identification. In our case, the problem lies in how to improve the ability to evaluate and prioritize green projects, knowing previously that the main problems and the development delay of the communities that belong to the post-conflict period in Colombia [6,15] are related to the availability and the use of the natural resources such as water and land [1,3]. That evaluation and prioritization capacity is understood as the selection of the most appropriate projects that can respond to the problems, needs, and expectations of the community [16,17].

The second step was the hypothesis planning, through which it was exposed that it is possible to anticipate where and at what moment of time to direct investments in green projects; keeping in mind, first, the availability and the use of the natural resources of water and land (supply and demand relationship), and in second place, citizen factors determined directly with the community through participation workshops.

In the third step for the creation of the modeling, we made the casual loop diagram or CLD (Figure 2) [18,19], in which we identified the relationships between state variables (supply and demand for water; supply and demand for land; and, of course, population), and the different parameters, among them the citizen factors. The fourth step was the creation of the stock and flow diagram or SFD (stock and flow diagram, Figure 3) [20,21], in which the casual loop diagram is complemented by secondary variables and parameters in order to simulate the behavior of the system over time.

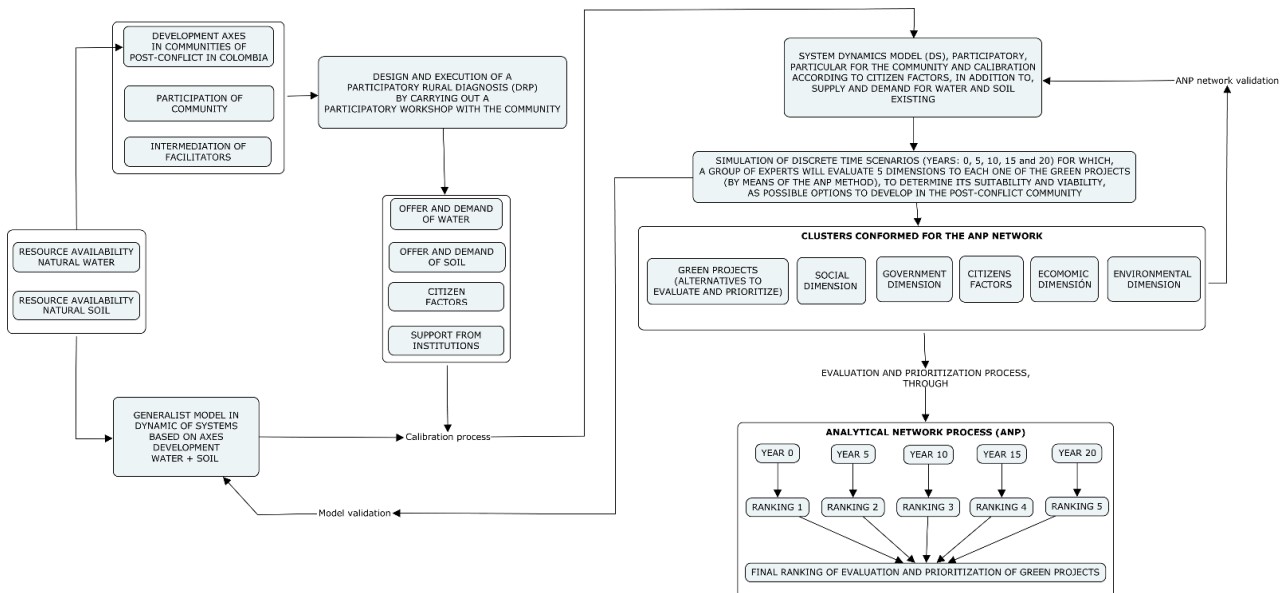

**Figure 1.** Methodological development for the evaluation and prioritization of projects.

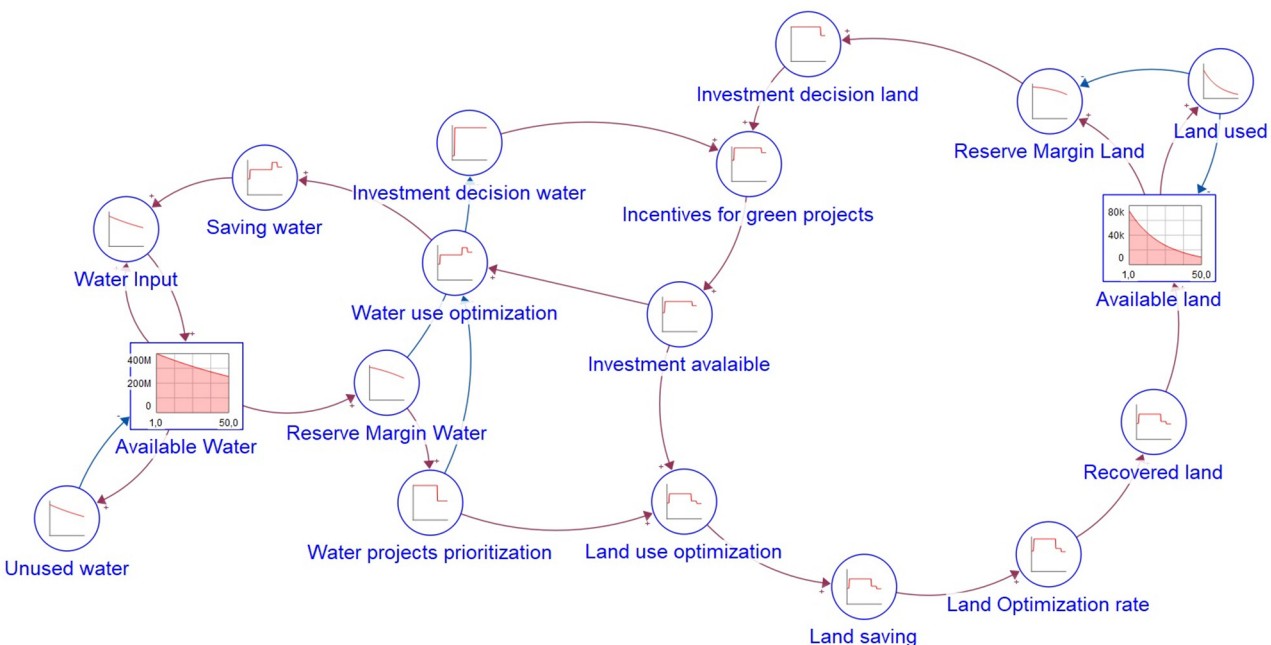

**Figure 2.** System dynamic model—casual loop diagram.

## STOCK AND FLOW DIAGRAMAM - SDF

**Figure 3.** System dynamic model—stock and flow diagram (SFD).

## 3. Validation of the Model Using System Dynamics

There are three cases of sensibility analysis for the models based on system dynamics: numerical sensitivity, mode of behavior sensitivity, and policy implementation sensitivity. After defining the model that will be applied to generate scenarios for the evaluation and prioritization of green projects, it is very interesting to explore the numerical sensitivity because with this it is possible to show the adjustments that are required to be carried out in order to validate the hypotheses initially raised.

The numerical sensibility occurs when changes in the assumptions alter the numerical results. For example, if the water input rate in the model is changed, how the output is altered in the Available Water level variable will be observed. All the models show numerical sensibility [18] and it is possible to observe how a reduced set of scalar factors can characterize the multidimensional uncertainty in a summarized but exhaustive way [22,23]. The proposed model for this thesis shows two parameters that are of interest and that are directly related to the level variables, the water input rate in the Available Water level variable and the land use rate in the Available Land level variable. To carry out the numerical sensitivity analysis, a window was used for the water inlet rate between 0.0 and 0.45, a window for land use rate between 0 and 0.20, and the Monte Carlo method that is included in the simulation package Vensim Pro was used; Figures 4–9 show the results. Due to solution map being very big, the Vensim Pro software was configured to make at least 2000 executions of the algorithm so that the results were consistent and cover all possible numerical solutions for the parameter ranges. It is important to mention that, for the sensitivity analysis, the uniform random distribution was used by means of which a fixed variation of the parameters is guaranteed.

In Figures 4 and 5, the sensitivity analysis of the variables Available Water and Available Land manages to establish a range in which the dynamics of the system evolves and makes the projects that can be developed according to the assumed parameters feasible and those that were obtained with the help of the community in the participatory workshops were carried out. In this way, the model provides to the experts a first selection criterion and becomes a filter that can be exploited by decision makers during the development of the peer evaluations proposed in the analytical network process (ANP) and that will be shown in the next section. Once the projects are framed within this range, analysis efforts

can be concentrated on the other variables of the system that account for other important factors within the complete evaluation and prioritization process.

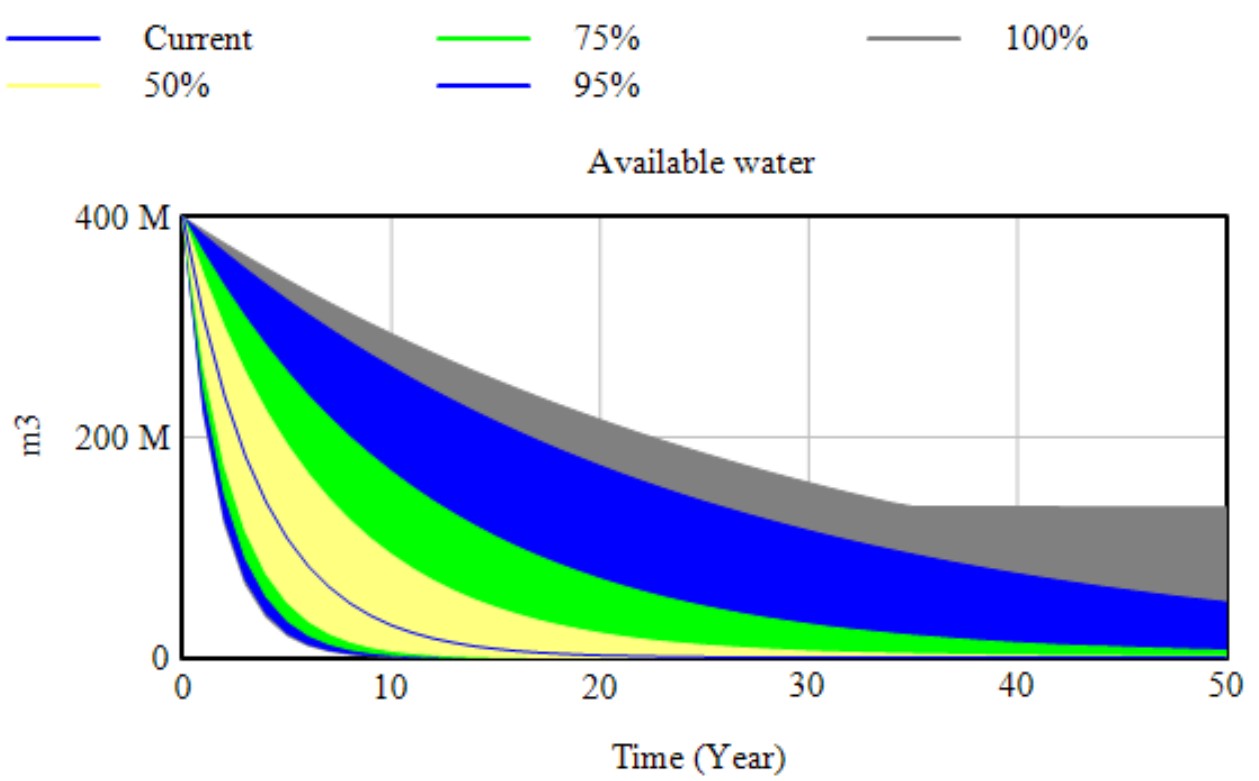

**Figure 4.** Evolution of the dynamic system according to sensitivity analysis of the Available Water level variable.

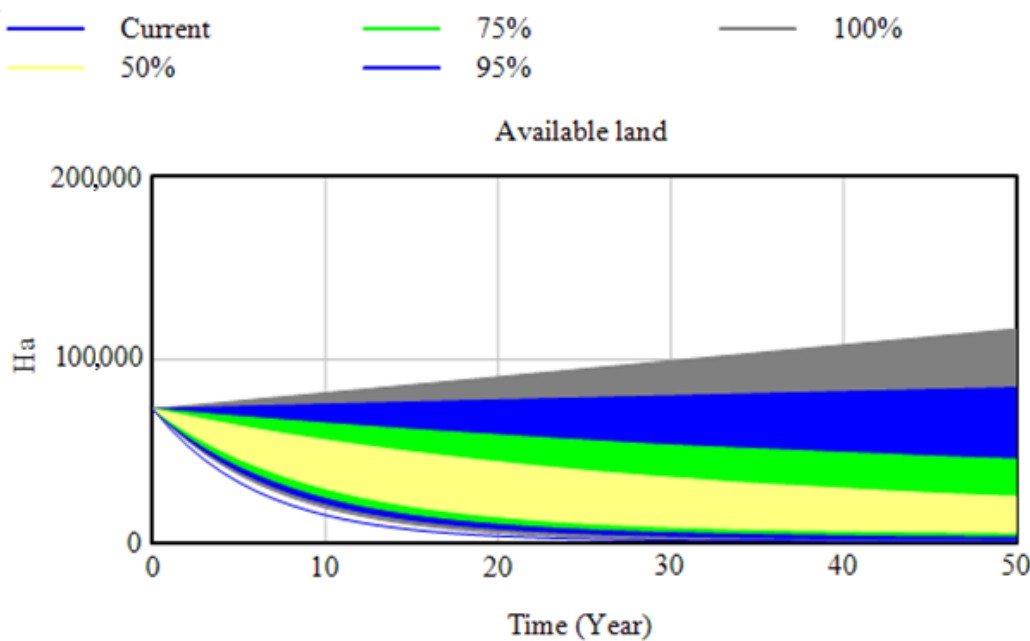

**Figure 5.** Evolution of the dynamic system according to the sensitivity analysis of the Land Available level variable.

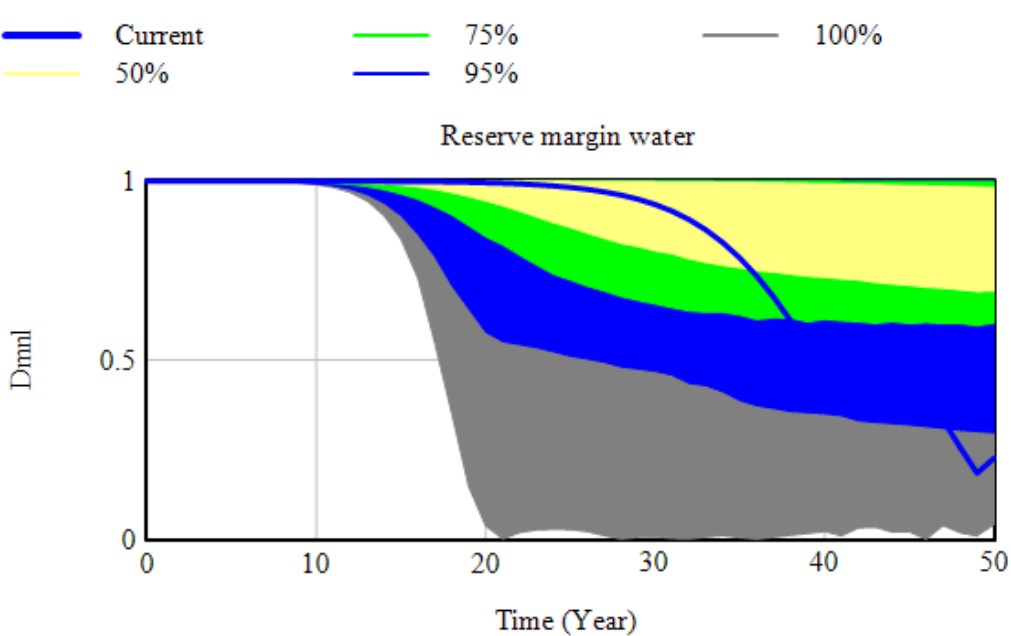

**Figure 6.** Behavior in the water reserve margin that indicates the availability over time of the natural resource for the projects.

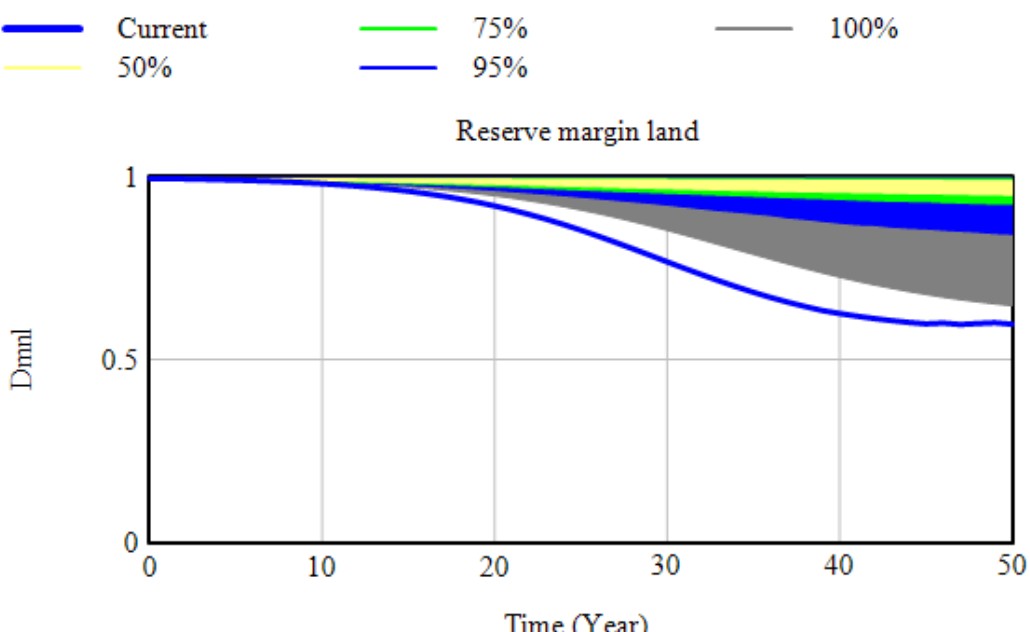

**Figure 7.** Behavior in the land reserve margin that indicates the availability over time of the natural resource for the projects.

The sensitivity analysis of the variables available water level and available ground level shows the percentage of accumulation of solutions in each of the bands that appear. For example, in the case of Figures 2 and 3, the blue curved line represents the solution business as usual, that is, the model solution is shown with the initial configuration unchanged. However, the blue, green, and yellow stripes represent the percentage of accumulation of solutions found for this case, the graph indicates that 95% of the results of the 2000 simulations are in the blue band, 75% of results are in the green band, 50% of the results are in the yellow band, and the gray stripe indicates the possible scenario of all simulations, that is, 100% of the simulations are located from the gray band to the other bands that overlap it.

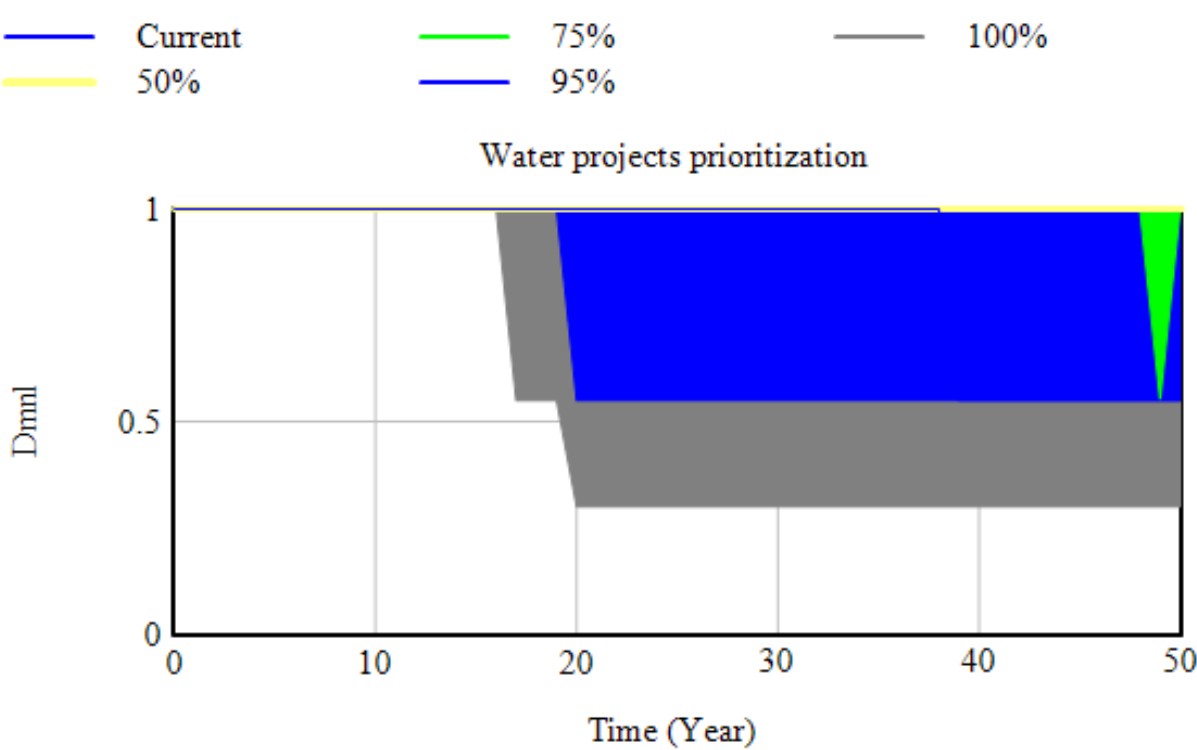

**Figure 8.** Monotonous behaviors in the water prioritization variable due to the influence of the citizen factors.

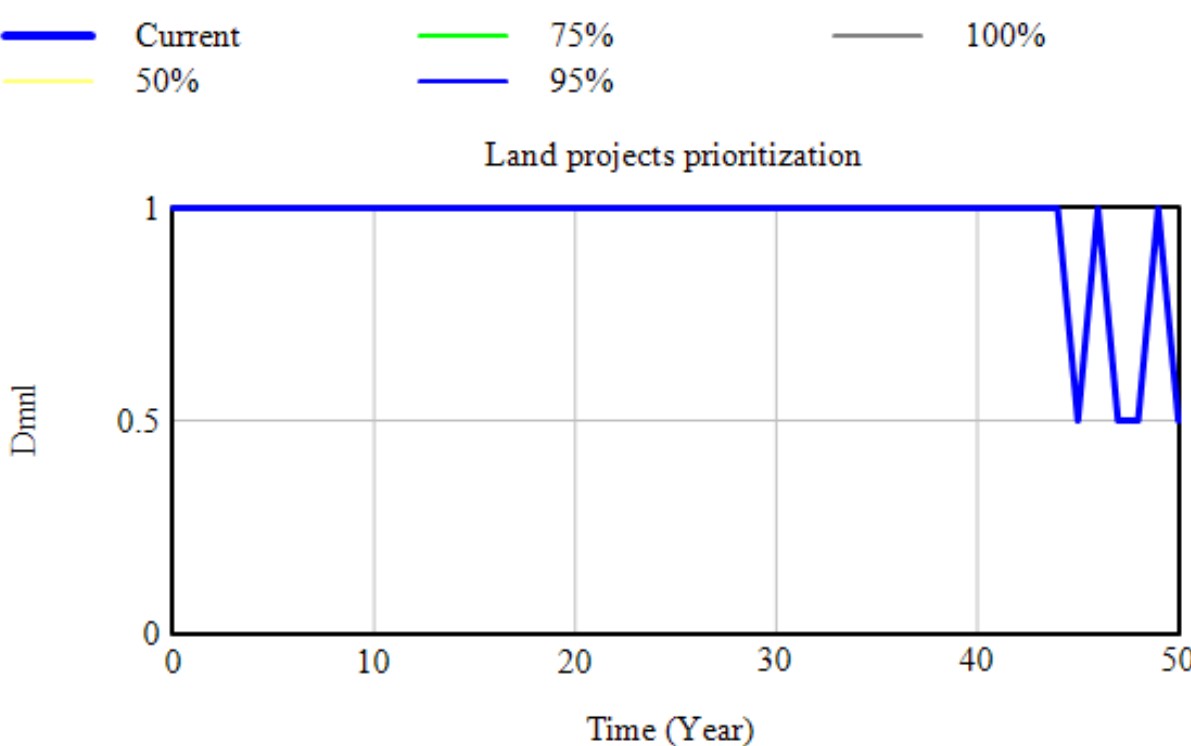

**Figure 9.** Non-scarcity scenario according to the land prioritization variable.

Now, in Figures 6 and 7, the sensitivity analysis focuses on the variables called reserve margins, both water and soil. This time the sensitivity shown by these variables accounts for the narrow margin that exists between the use and abuse of natural resources of water and land, which, in turn, can mean their abundance or scarcity. Therefore, this analysis

could also guide investment decision-making regarding one project or another, depending on the community and depending on the availability of resources in the area.

As we said previously, in Figures 6 and 7 the sensitivity analysis focuses on variables called reserve margins, both water and soil. This time, the sensitivity shown by these variables shows the narrow margin that exists between the use and abuse of natural resources of water and land; at the same time they can show the abundance or scarcity. Therefore, that analysis could also guide investment decision-making regarding one project or another, depending on the community and depending on the availability of resources in the area.

For the case of Figure 6, the simulation indicates that at some point there could be a zero margin, but not a negative margin. This statement coincides with the dynamic hypothesis raised in the model from the beginning, where it is not considered that there may be missing water. However, if this case occurs in the sensitivity analysis, what could be assumed is that there is an unattended demand or perhaps an excess demand for the resource.

However, as we explained previously, 95% of the simulations are in the blue band and that stage shows us that only 5% of the solutions would lead to a shortage. For the rest of the cases, the sensitivity analysis shows a scenario under which the reserve margin has a limit between 30% and 100% approximately, according to the conditions.

On the other hand, when the project prioritization variables are analyzed, Figures 8 and 9, a lower degree of sensitivity is evidenced, that is, more monotonous behaviors in the system, because such variables are directly influenced by the factors citizens. However, everything will depend, again, on the availability of natural resources in the area at the time of evaluating and prioritizing the projects, and the problems and needs of the community that can be quantified in the citizen factors, with which the model is calibrated.

Additionally, 75% of the simulations are in the green band. This indicates that between 95% and 75% of the solutions found, the reserve margins show scenarios under which there is no shortage of supplies compared to the current policy; that is, with the conditions that were raised, the model has a behavior scenario where all the solutions will move within that margin. Therefore, if the sensitivity range is restricted or moved, the gap or scenario of possibilities should be closed.

*Exploitation of the Model*

Validation techniques tend to identify whether the model that is available is faithfully coupled to the real system or phenomenon that is being analyzed. However, when behavior patterns change as model assumptions change, we talk about sensitivity in the way of behavior [24]. These changes in assumptions can occur during the model optimization process, when it is necessary to modify a pattern that has been diagnosed as inconvenient or negative. Faced with the new alternatives, the model could go from an oscillatory behavior to a monotonic behavior. At this point, it is then also possible to use the tools that come with software such as Vensim Pro, Stella Architect, or any other type of software based on system dynamics, in order to obtain a more rigorous analysis of the system.

In short, it is possible to use these tools in models that can be represented by systems of ordinary differential equations and that additionally present no smoothness or jumps in their functional expressions. The tools provided by Vensim Pro and Stella Architect allow to show some phenomena associated with the use of stepped functions and thus evaluate alternatives and additional information in order to exploit the model in a better way.

For the model shown in the Figure 3, the auxiliary variables Optimization of Water Projects and Optimization of Land Projects are step functions, which in turn depend on the Reserve Margins (water and land, respectively) which are equally non-smooth. Hence, it is of great interest to evaluate what happens with investment decisions and the optimization of water and land use, when citizen factors are altered. In Figures 10 and 11 we can see that for each value of each one of the citizen factors in the interval between 0 and 1, the output of the variables Water Projects Prioritization and Land Projects Prioritization shows

a variation of less than 50% during the first 30 years, but varies substantially during the last 20 which is the time in which it is most affected by the alteration of the variables related to the reserve margins (MRA and MRS) which are very important at the time of evaluating and prioritize possible projects to be carried out. In this case, projections are made on these auxiliary variables to understand how they are affected by specific values in the parameters of citizen factors. The greater variation at the end of time implies changes in investment decisions, which would lead decision makers to move towards other types of investments, such as social or environmental, that respond to the conditions of the moment. With this tool given by Vensim Pro sensitivity two types of behavior are examined, that is, for values around 0 the dynamics system is monotonous, meanwhile for values above 0.5 oscillations occur. It is worth mentioning that these behaviors are within the possible intervals that a rural community would face in a post-conflict period, which is characterized by a monotonous and slow behavior associated with the speed of development. Therefore, for decision makers regarding the evaluation and prioritization of green projects, it is very important to know what happens before a change in the parameters in the medium and long term and is not normally undertaken when studying dynamic systems in their stationary state

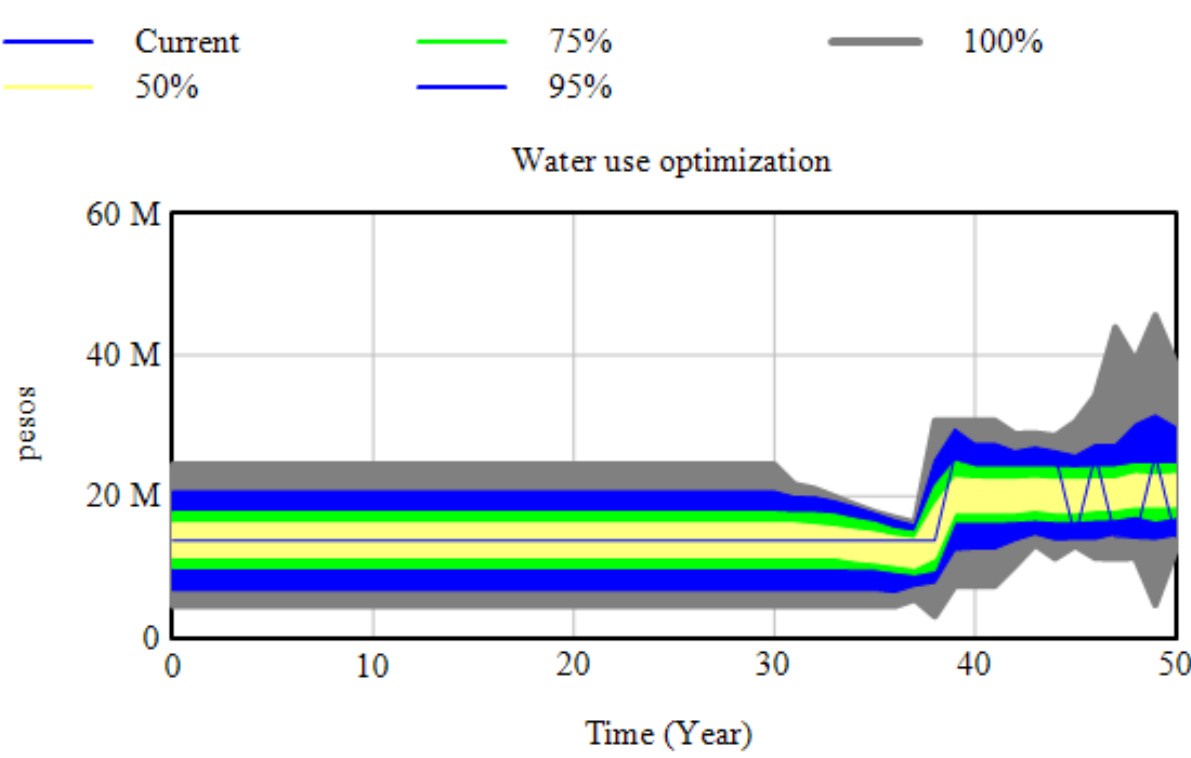

**Figure 10.** Variation over the time for the variable Water Use Optimization.

To finish model validation, a policy sensitivity analysis is presented that consists of making changes in the hypotheses in order to reverse the impacts or convenience of a proposed policy. For example, in the case of the model for the evaluation prioritization of green projects, it is proposed to change the diagram under which investment decisions are calculated (water and land), analyzing two possible scenarios.

In the first test model, to calculate the value of the variable Incentives for Green Projects, a greater weight is given to the variable Investment Decision Water maintaining no changes in the variable Investment Decision Water, and a greater weight is given to the variable Investment Decision Land. Now, exploring extreme cases, a second test model is proposed, in which one of the variables could be Investment Decision Water or Investment Decision Land becomes zero, in order to observe the changes in the variables Water Use Optimization and Land Use Optimization.

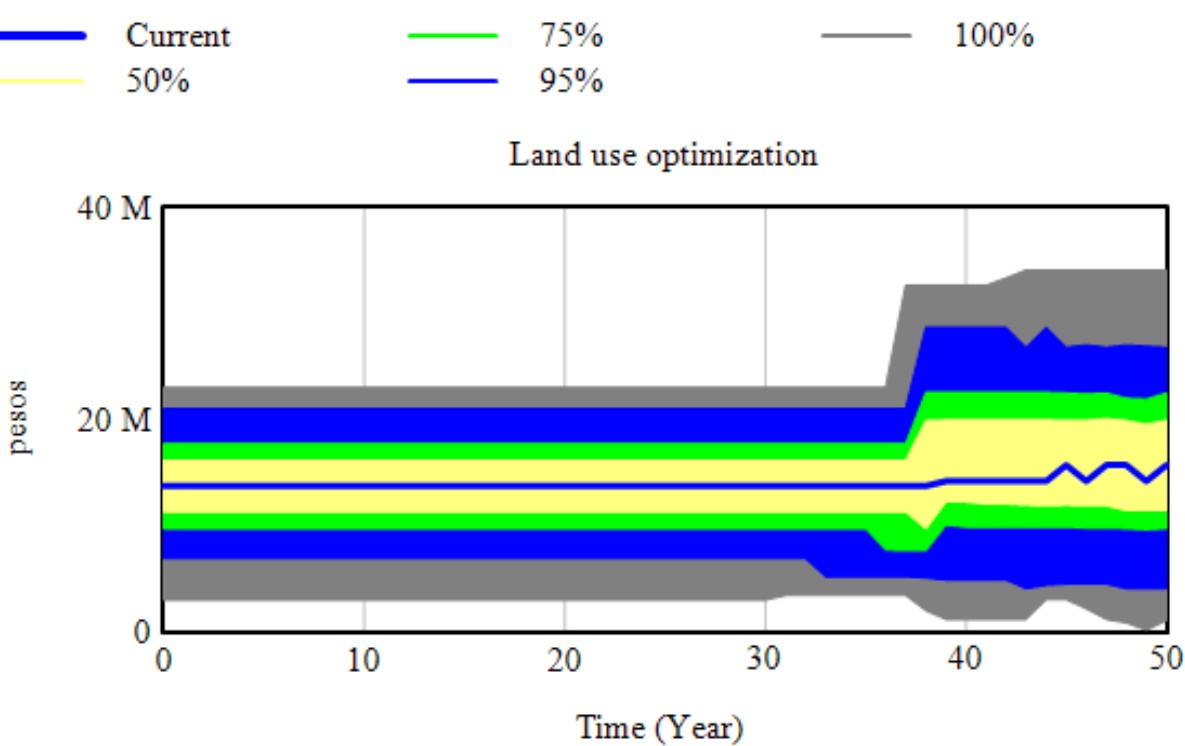

**Figure 11.** Variation over the time for the variable Land Use Optimization.

When carrying out the sensitivity analysis before the policies of the diagrams in the first, second, and third test models, it can be seen that when the weight varies in the variable Investment Decision Water or Investment Decision Land, for the first test model, it presents greater variability than the second and third test models, see Figures 12–15.

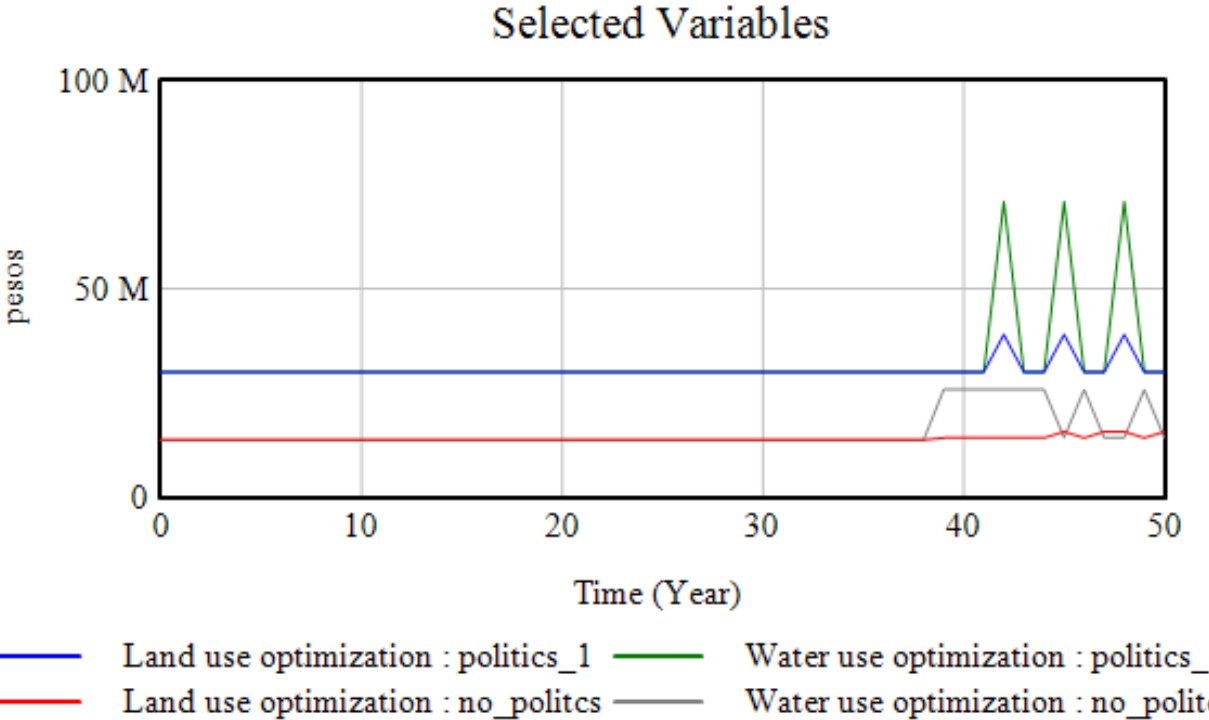

**Figure 12.** Scenario without investment policies versus a scenario with investment decisions.

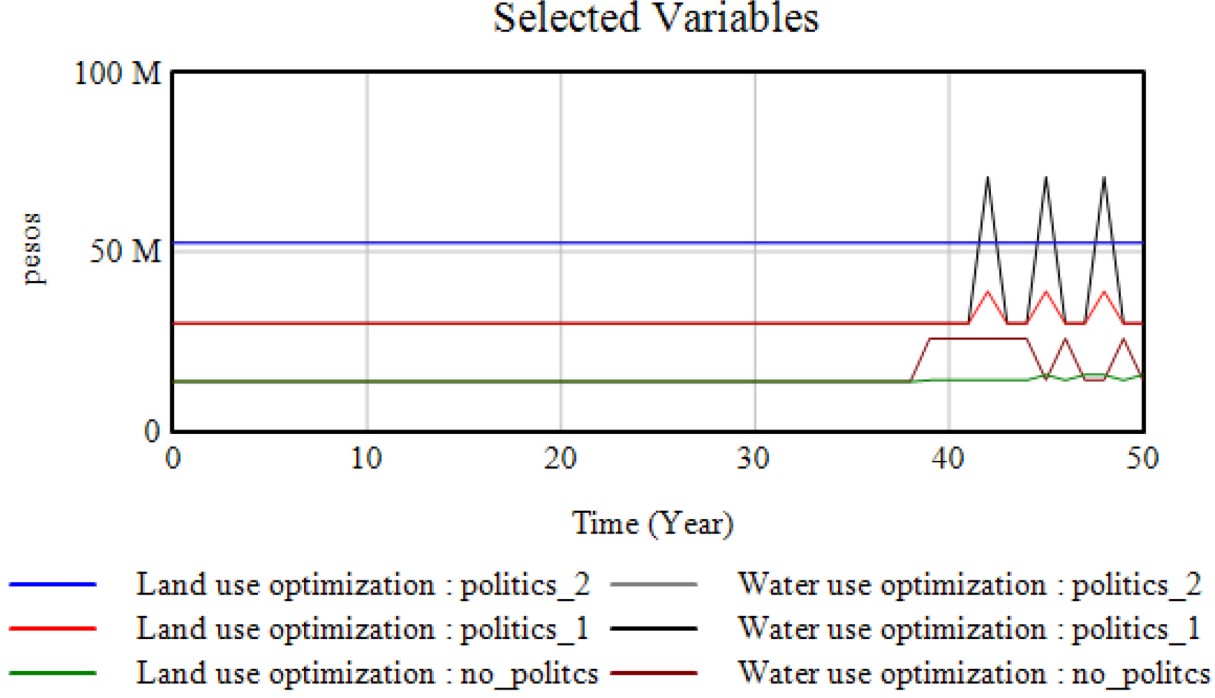

**Figure 13.** Second test model—variables Land Use Optimization and Water Use Optimization.

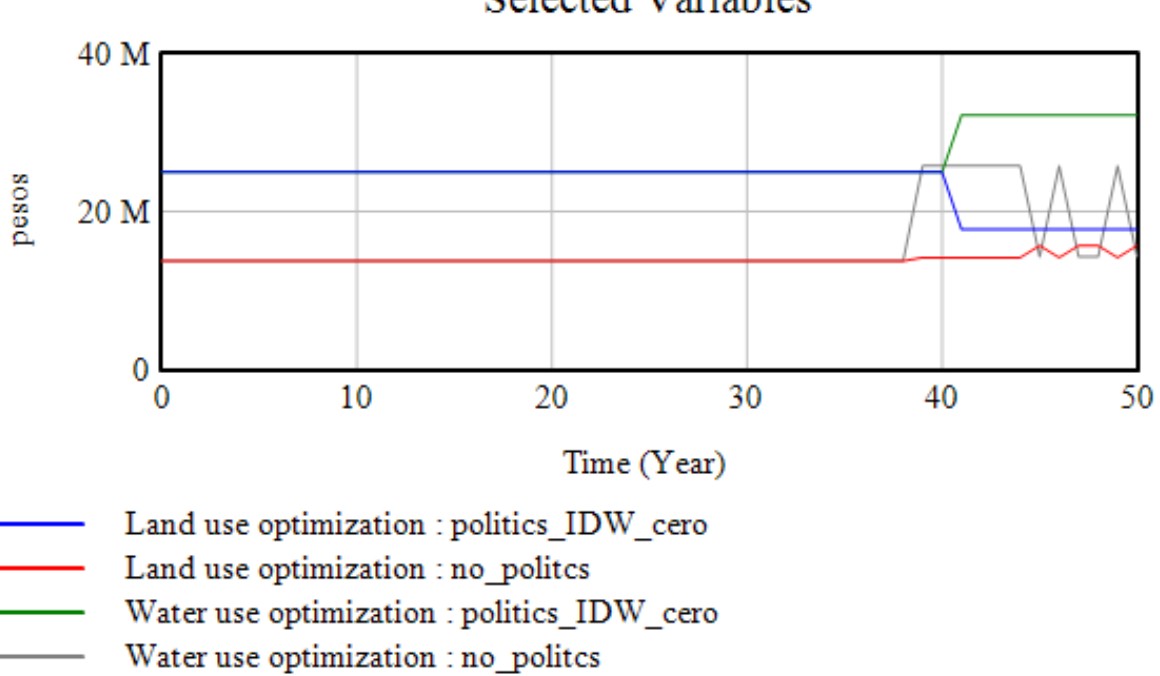

**Figure 14.** Third test model—Investment Decision Water (Zero), reduction in the variability of decisions thanks to citizen factors.

If it is possible to establish the weights of the variables regarding the investment decision for which the maximization of the variable incentive for green projects is achieved, the dynamic flow of this system evolves without having to change between investment and non-investment decisions. In this way, the variability in the decisions of the entities in charge of evaluating and prioritizing the projects for this community is reduced. Additionally, if we compare the variables Investment Decision Water and Investment Decision Land

(weighted), with the variables Water Project Prioritization and Land Use Optimization, it is clearer that the parameters associated with these, that is, the citizen factors, are the ones that significantly affect the behavior of the system. Therefore, sensitivity analysis efforts can be concentrated on them to obtain more out of the model.

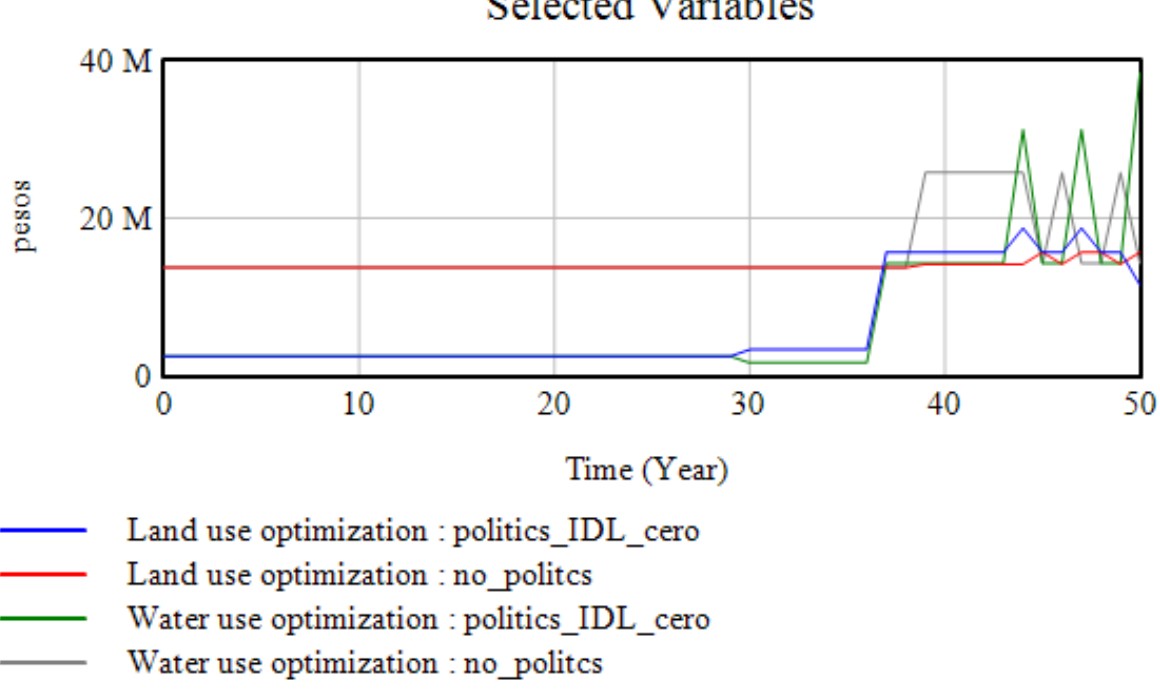

**Figure 15.** Third test model—Investment Decision Land (Zero), unfavorable conditions for resource optimization.

In conclusion, according to the results of the validation it is possible to say that the model is valid because in front of a variation in parameters in the specific ranges determined above, it is shown that all the simulations used for the sensitivity analysis respond consistently with the dynamic hypothesis proposed for this research.

## 4. ANP Network Structure

The second phase of the proposed methodology corresponds to the evaluation and prioritization of projects through the multi-criteria decision method ANP (network analytical process) [7,25,26]. The ANP method allows to identify and consider mutual dependencies between the different evaluation criteria [13]. Therefore, it is possible to structure a cluster and node network and, therefore, define the interrelationships that occur between them [27]. The general structure of a network ANP is composed of source nodes and destination nodes (drain) that are determined when the paths of influence are established [28]. The connections between nodes and elements can occur through feedback to components of other elements by ringlets to the same components [29]. Due to the large number of interdependencies and interactions in the proposed system dynamics model, likewise, the impossibility of structuring it hierarchically, we decided to use the ANP method like the method for the evaluation and prioritization of the projects (alternatives). The creation of the structure of the ANP network started from the system dynamics model and was carried out in different steps.

The first step consisted of the identification, in the model in DS, of the clusters that would make up the ANP structure. At this point we identified six clusters that corresponded to the social, environmental, and economic dimensions. In addition to the citizen factors and the government dimension, the ANP structure was completed with the cluster that belongs to the alternatives or projects to be evaluated and prioritized, as shown in Figure 16.

**IDENTIFICATION OF THE CLUSTERS OF THE ANP NETWORK IN THE SYSTEM DYNAMIC MODEL**

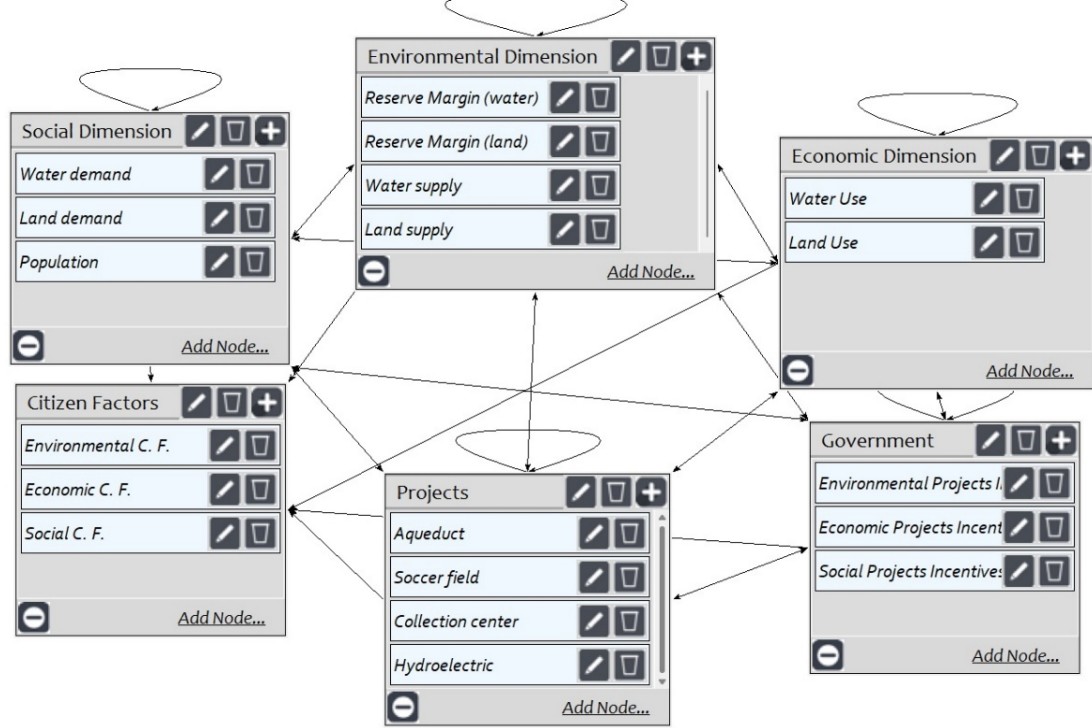

**Figure 16.** Identification of the clusters of the ANP network, in the system dynamics model.

The second step consisted of the identification of the nodes and elements that make up each cluster. In this way, those variables and parameters of the model that were found were established as elements inside each identified cluster in the previous step. Likewise, the interconnections and dependency relationships were established according to the existing relationships in the model in DS. In Figure 17 we can see the complete structure of the ANP network.

**Figure 17.** ANP network from the system dynamics model.

## 5. Study Case California County (Fresno, Tolima)

The application of the proposed methodology was carried out through a case study in the county California, located in the Tolima department in Colombia (see Figure 18). Therefore, this zone belongs to the region of the downtown of the Magdalena region historically affected by the Colombian conflict until 2016 in which the peace agreement was signed. The county extends over 150 hectares and is located on the eastern slope of the Central mountain range at an elevation of 1800 m above sea level and belongs to the Guarinó river basin, an important tributary of the Magdalena river, one of the most important rivers of the country.

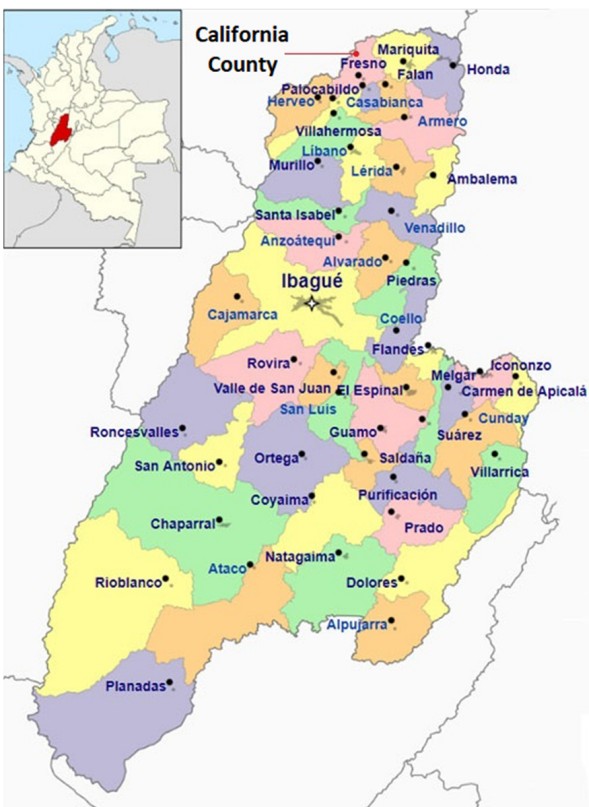

**Figure 18.** Geographic location of the California county, Fresno (Tolima); source: Tolima Governing [30].

### 5.1. Participatory Workshop (Participatory Rural Appraisal—PRA)

Through the participatory workshop carried out with the community of the county, it was possible to gather the necessary information to calibrate the balance model in system dynamics, according to the variables and parameters proposed related to the supply and demand of water and land; likewise the citizen factors and support from local and national government, as well as other funding institutions. The workshop with the community was used the method called participatory rural diagnosis [11]. Table 1 shows the main findings, classified according to the dimension (clusters of the network ANP) to which they correspond (social, environmental, economic, and government). Therefore, the citizen factors that were determined by the facilitating team of the workshop were presented after finishing it and the alternatives, that is, the different projects to be evaluated and prioritized which also belong to the six clusters of the network. Figures 19 and 20 show images of the workshop in the community hall.

**Table 1.** PRA findings classified by dimension (ANP network cluster).

| Identified Finding | Dimension (ANP Network Cluster) | Used Tool |
|---|---|---|
| Difficulties in arriving to the county | Socioeconomic | Venn diagram |
| Small-scale agriculture (self-supply and external sale) | Socioeconomic | Venn diagram |
| Main products: cocoa, banana, avocado, and coffee | Socioeconomic | Venn diagram |
| Climate variety, many farming and fertile lands | Economic | SWOT |
| No agro-tourism-type projects are identified in the region | Economic and from the government | SWOT |
| Social connection | Social | SWOT |
| Abandonment by the state and local government | Social and from the government | Venn diagram |
| The population of the county, most of them left the county due to the violence in the nearby regions | Social | Semi-structured dialogue |
| Diversity of thought and approach to the problems and needs of the community | Social | Semi-structured dialogue |
| Impossibility of taking adequate advantage of natural resources and the existing share capital due to the lack of institutional support in the development of high-impact projects and lack of training for its inhabitants | Socioeconomic and from the government | Semi-structured dialogue |
| Soil affectation (in exchange for better productivity) using pesticides and agrochemicals in crops | Socioeconomic | Semi-structured dialogue |
| The community of California recognizes that technification in their crops can mean greater competitive capacity in the market, and a better position of their products with more profitable prizes | Socioeconomic | Semi-structured dialogue |
| The natural resource of water is widely accessible to the inhabitants, so it can be consumed directly from the springs and gorges of the zone | Environmental | Semi-structured dialogue |
| The community recognizes the problems that underlie the consumption of non-potable water in relation to the generation of diseases. | Social environmental | Venn diagram |
| Industrialization processes or widespread growth of the main crops are not identified in the county, small-scale agricultural production being the main economic source, likewise the underemployment in the nearby counties | Economic | Venn diagram |
| Project management or resources for productive projects in the county are not identified, which makes the improvement in the physical, economic, natural, and cultural conditions of the community difficult | Government | Semi-structured dialogue |

*5.2. Model Calibration in System Dynamics*

The calibration of the proposed model was made based on two main sources, the first corresponding to the participatory workshop developed with the inhabitants of the county California and the second corresponding to secondary sources such as the web pages of the mayor's office of the county and from foundations and private corporations that work in the region supporting the community through rural development projects, among others. As a result, Table 2 is presented below in which the parameters, their value, and the source from which it was taken are described. Likewise, the table shows the value assigned to each citizen factor and other assumptions that are considered within the system.

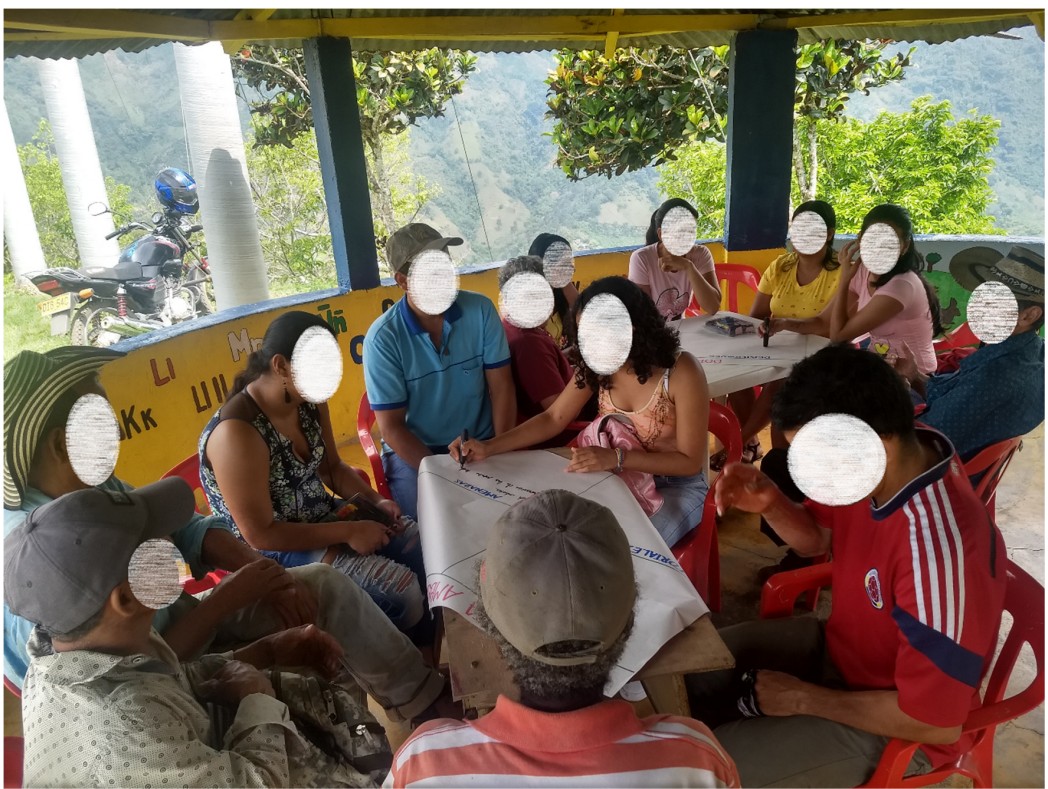

**Figure 19.** Development of the SWOT matrix during the participatory workshop with the California community.

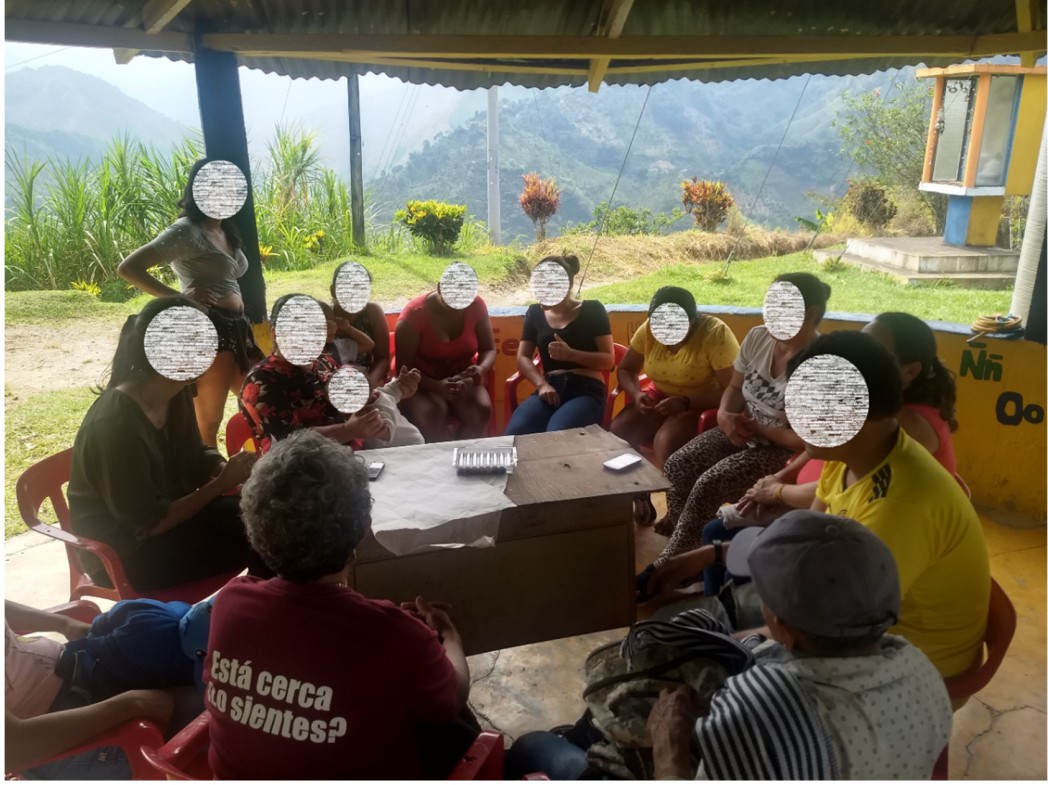

**Figure 20.** Semi-structured dialogue activity developed with workshop participants in California county.

**Table 2.** Calibration of the model in system dynamics.

| Parameter | Description | Value | Units | Source |
|---|---|---|---|---|
| Water Input Rate | Amount of water provided by the different tributaries in the area per year | 22 | % | [30,31] |
| Unused Water Rate | Amount of water not consumed or used per year | 45 | % | [30,31] |
| Variable Consumption of Water | Amount of water consumed above the annual average | 60 | Cubic meters | [30,31] |
| Fixed Consumption of Water | Amount of water consumed per year | 130 | Cubic meters | [30,31] |
| Population | Number of inhabitants | 200 | People | [30,31] |
| Price of Water | Value paid per $m^3$ of water | 6000 | Pesos | [30,31] |
| Available Budget | Amount of money available for investment | 50 | Millions of pesos | Model assumption |
| Price of Land | Average value paid per hectare of land | 150 | Thousands of pesos | [30,31] |
| Land Use | Amount of land consumed per year | 9000 | Hectares | [30,31] |
| Intended Use of Land | Proportion of land (of the total) planned to be used for the following years | 30 | % | [30,31] |
| Land Use Rate | Proportion of land (of the total) used each year | 15 | % | [30,31] |
| Total Land | Total hectares of land (used and unused) | 515 | Thousands of hectares | [30,31] |
| $FCS_{water}$ | Social citizen factor (water) | 20% | Dimensionless | Participatory workshop |
| $FCA_{water}$ | Environmental citizen factor (Water) | 25% | Dimensionless | Participatory workshop |
| $FCE_{water}$ | Economic citizen factor (water) | 55% | Dimensionless | Participatory workshop |
| $FCS_{land}$ | Social citizen factor (land) | 10% | Dimensionless | Participatory workshop |
| $FCA_{land}$ | Environmental citizen factor (land) | 30% | Dimensionless | Participatory workshop |
| $FCE_{land}$ | Economic citizen factor (land) | 60% | Dimensionless | Participatory workshop |

*5.3. Model Simulation for Evaluation and Prioritization*

Next, a series of simulations of the proposed model are presented that were carried out using the Stella Architect software. In these, the variation in the different variables of interest in the model can be observed which were presented to the experts to be considered in the evaluation and prioritization of green projects. Once again, as was the case in the study carried out with this same methodology in Pesebre county (Tame, Arauca) [6], the hypothesis that was raised from the beginning is confirmed. First of all, the reserve margins (water and land) of Figures 21 and 22 show how the change in the levels of supply and demand for water or land, represented in the defined margins, generate early alerts which allow adequate control of investments.

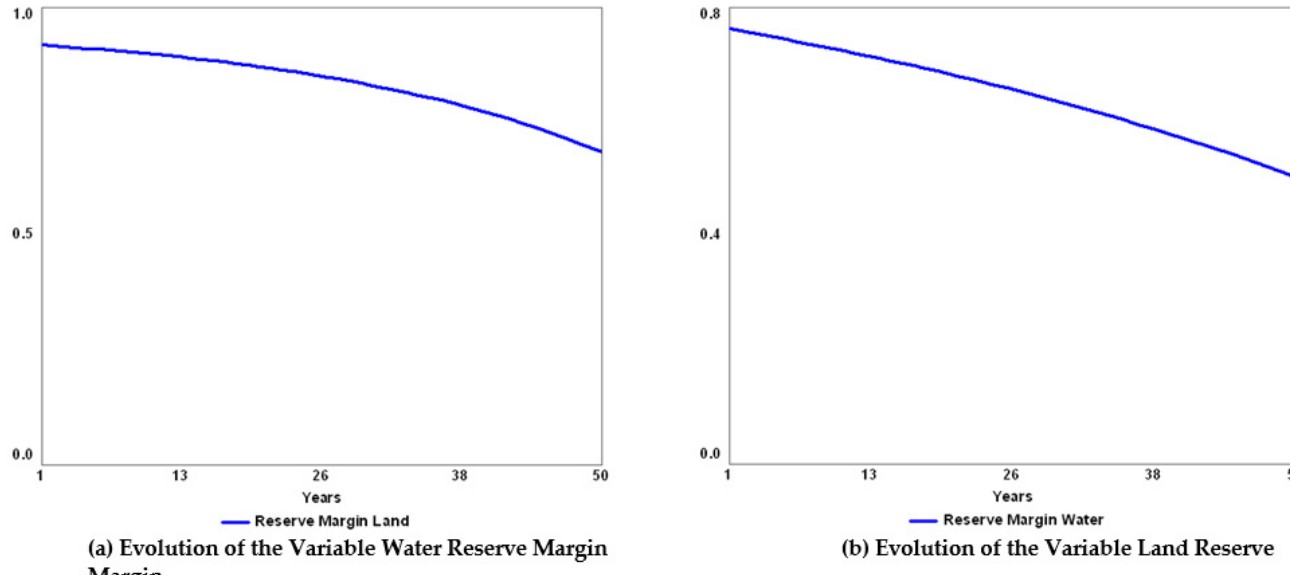

**Figure 21.** Simulation reserve margins environmental dimension.

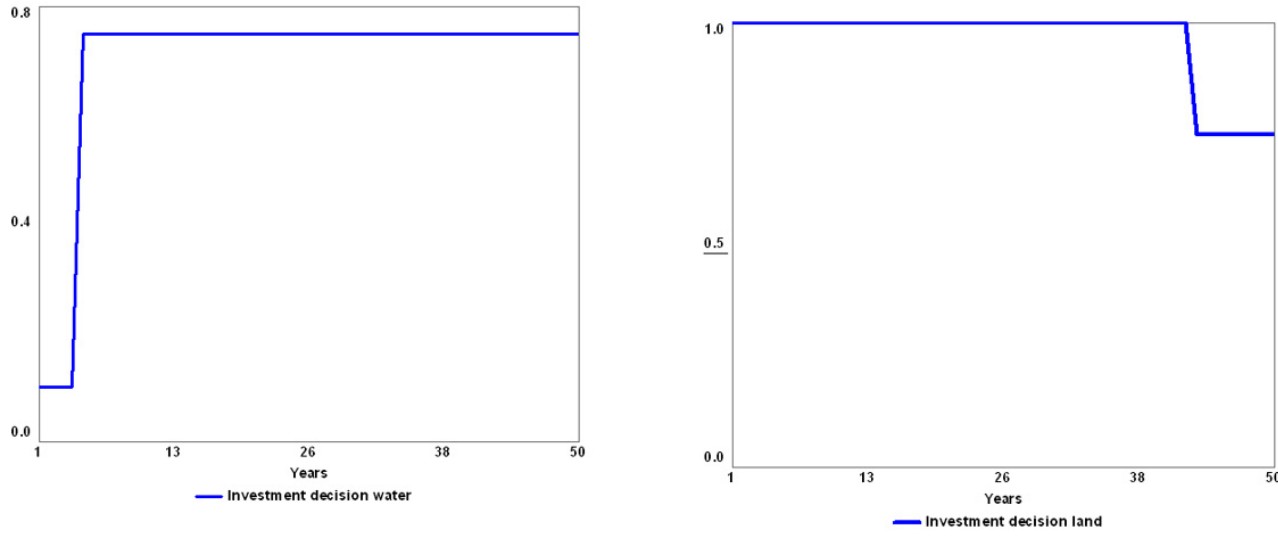

**Figure 22.** Feasibility of the projects according to the reserve margins and the citizen factors (government dimension).

Secondly, we can see how the variables related to the prioritization and optimization of water and land resources are also influenced by citizen factors, established through community participation. In accordance with Figure 23, for the case of California county, it is evident how the community gives preference to water projects, that is, those in which their water sources can be used for a social purpose for their inhabitants.

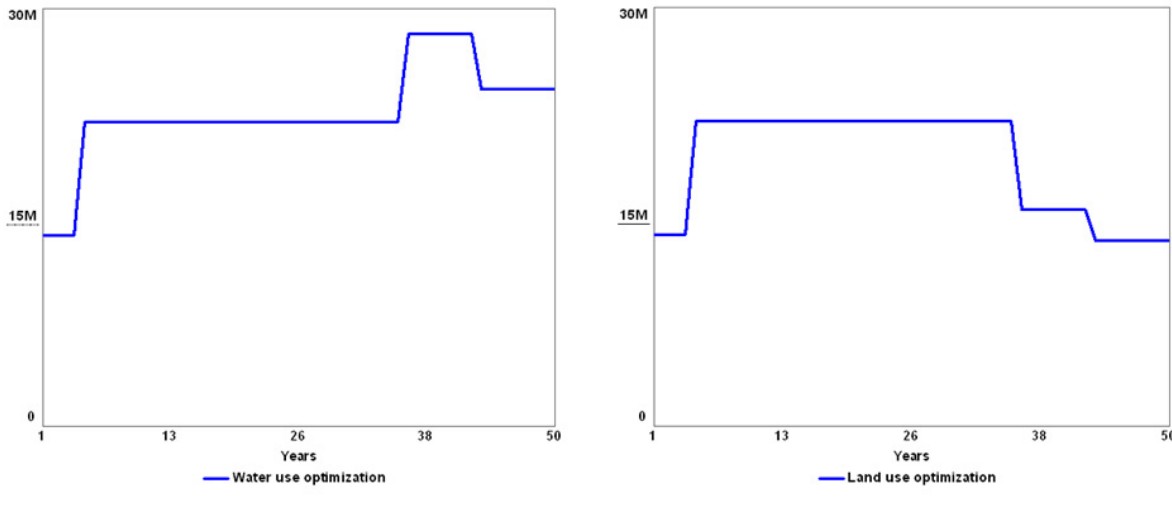

(a) Water use optimizarion        (b) Water use optimizarion

**Figure 23.** Optimization of the resource economic dimension.

### 5.4. Project Evaluation and Prioritization with ANP

Once the simulations were carried out with the model in system dynamics (DS), the second phase of the proposed methodology began that consisted of the application of the analytical network process (analytic network process—ANP). Having structured the ANP network, which was described in Section 3, the six dimensions proposed (social, environmental, economic, government, citizen factors, and projects), the elements (variables), and their corresponding elements (variables) were evaluated according to dependency relationships and defined feedback.

As mentioned in Section 3, the ANP method is based on pairwise comparisons [32,33] to determine the priorities among the indicators or variables involved in the evaluation. The experts were asked to, according to the information presented from the model in system dynamics, evaluate the relevance of the indicators considered according to the fundamental scale of Saaty [34] (Table 3).

**Table 3.** Saaty fundamental scale.

| Value | Definition | Comment |
|---|---|---|
| 1 | Equal importance | Criterion A is just as important as criterion B |
| 3 | Moderate importance | Experience and judgment slightly favor criterion A over criterion B. |
| 5 | Big importance | Experience and judgment strongly favor criterion A over criterion B |
| 7 | Very big importance | Criterion A is much more important than criterion B. |
| 9 | Extreme importance | The greater importance of criterion A over B is beyond doubt |
| 2, 4, 6, 8 | Intermediate values between the previous ones, when it is necessary to qualify | |

The first step of the evaluation refers to the pairwise comparisons between the clusters [27,35], that is, among the six dimensions defined. Once the evaluations were compiled it was possible to develop the evaluation matrix [36,37], in which the numerical values represent the influence identified for the elements of the network. The second step after having established the priorities of the clusters was to perform pairwise comparison for the nodes or elements or each cluster [38,39]. Regarding the evaluation of each cluster, the

judgments were made considering the influences and interdependencies [12,40] recognized in the network. For example, Figure 24 shows the pairwise comparison between "social dimension" and "economical dimension". In detail, the assigned value "4" refers to the fundamental scale of Saaty (Table 3). This means that the experts considered the "Social Dimension" more important than the "Economic Dimension".

**Figure 24.** Pairwise comparison. Environmental dimension vs. economic dimension.

Figure 25 shows the pairwise comparison between the nodes (elements) that belong to environmental dimension (cluster). Undertaken in the same way as the pairwise comparison by the clusters for the assignment of values, the fundamental scale of Saaty was used in order to determine the importance of the different variables. In detail, the assigned values mean that, first, the "water reserve margin" is more important than "the land reserve margin" (value 3); second, the "water offer" is equally or moderately more important than "land offer" (value 2); and third, that the "water reserve margin" is moderately more important than the "land supply" (value 3). Great importance was given to the "water reserve margin" in the county California. This is due in part to the fact that water is of vital importance for crops and thanks to the local aqueducts all the inhabitants can benefit from it. On the contrary, regarding water or land, which is the private property of the inhabitants of the community or of companies that established their crops in the area, since the benefit is individual it is taken advantage of. In this context, it is possible to underline that the comparison by pairs was carried out considering the characteristics of the area, and likewise the problems and needs of the California community.

**Figure 25.** Comparison between nodes with the Saaty scale.

Once all the pairwise comparison matrices were obtained for all the dimensions and the nodes, we developed the unweighted supermatrix [41,42], in which all the priorities obtained through pairwise comparisons are detailed. Likewise, the supermatrix represents the relations between the nodes that compose the network. Now, to obtain the weighted supermatrix, it was necessary to multiply the unweighted matrix by the final vector of priorities [43,44]. It is important to highlight that both for the construction of the ANP

network and for the peer evaluations and the development and operation of the matrices we used the super decisions software that was created by the foundation creative decisions [45].

## 6. Results and Optimization

As was expressed in the previous sections, the ANP method was carried out at five different instants of time and considered the information obtained by means of the system dynamics model for each moment of time. In this way, it was possible to observe the various interpretations that the panel of experts can make when there are many data. At this point, the usefulness of a multicriteria process is highlighted like the ANP, through which it is possible to group different decisions about the same issue and establish a valid solution [46].

Figure 26 shows the individual results of the network analytical process in each of the five moments that the evaluation and prioritization were carried out. For this research, a 20-year time horizon was used, the year 0 (without intervention) being the first year to be evaluated, followed by the years 5, 10, 15, and 20 as the last year to be considered. This case corresponds to the California county in the Magdalena downtown region, a Colombian post-conflict period community.

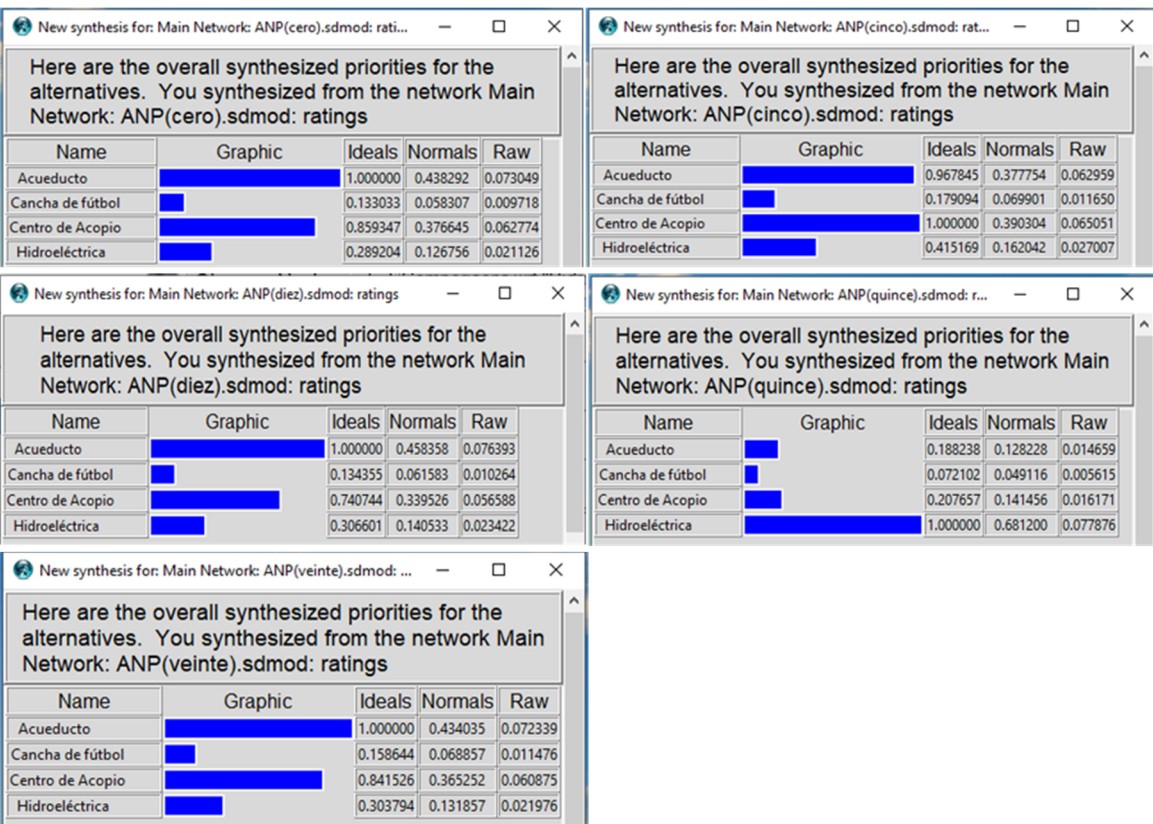

**Figure 26.** Results of the ANP evaluations (years 0 to 20).

However, the optimization process is based on the proposal described by Chichilinsky [47], to achieve sustainability. The basic idea is to exhibit a trade-off between preferences for the future and underlying natural resources (future supply and demand) and present preferences and consumption generated by utility criteria (current supply and demand). For this research, it was decided to give greater weight to the first ANP evaluations, 40% to the first one (year 0), 30% to the second (year 5), 15% to the third (year 10), 10% to the fourth (year 15), and 5% to the last evaluation (year 20). Considering that citizen factors express the most pressing needs and they are a Colombian post-conflict period community, its inhabitants expect a response in the short- and medium-term that improves their quality of life. Table 4 shows the results of each ANP evaluation realized and its corresponding weight.

**Table 4.** ANP evaluations and weights (years 0 to 20).

| Project | Year 0 | Weight 40% | Year Five | Weight 30% | Year Ten | Weight 15% | Year Fifteen | Weight 10% | Year Twenty | Weight 5% | Final Result |
|---------|--------|-----------|-----------|-----------|----------|-----------|--------------|-----------|-------------|-----------|--------------|
| Aqueduct | 0.438 | 0.175 | 0.377 | 0.113 | 0.458 | 0.068 | 0.128 | 0.012 | 0.434 | 0.021 | 0.391 |
| Soccer Field | 0.058 | 0.023 | 0.069 | 0.020 | 0.061 | 0.009 | 0.049 | 0.004 | 0.0688 | 0.003 | 0.061 |
| Collection Center | 0.376 | 0.150 | 0.390 | 0.117 | 0.339 | 0.050 | 0.141 | 0.014 | 0.3652 | 0.018 | 0.351 |
| Hydroelectric | 0.126 | 0.050 | 0.162 | 0.048 | 0.140 | 0.021 | 0.681 | 0.068 | 0.1318 | 0.006 | 0.195 |

## 7. Discussion

The proposed methodology for the evaluation and prioritization of green projects presents the use of applied mathematics [25,37,48], in a field in which it has rarely been used, such as sustainable social development [49,50]. Although the authors of [14] or [51] present an integrated approach to system dynamics and the ANP, this research could be more important considering the needs of the community with the goal of finding a balance between social, environmental, and economic development as set out in [52,53]. Although it is true that contributions such as those found in [25] in the field of sustainability are novel, they refer to other topics such as resource conservation, risk analysis [54], intervention strategies [55], among others, so the proposed methodology constitutes a contribution to the science of decision making and the integration of mathematics with the social and environmental aspects of the world so that, regardless of any bias or influence of sectors or people, the benefits to the community are maintained as we can see in [56,57].

Furthermore, in places where environmental conditions are often very favorable in relation to the supply of natural resources such as water and ground, but social and economic conditions require a more detailed review, the proposed methodology distances itself from research such as that presented in [58,59], by integrating different methods and having a model that is calibrated according to participatory factors (citizen factors), ensuring that the system includes not only the problems and needs but also the community expectations. Therefore, it gives rise to social learning processes through which it is possible to characterize the community as exposed in [9,56]. Moreover, just like the investigations presented in [60,61], the proposed methodology also achieves the characterization of the environment, information that is used when evaluating and prioritizing, since when applying a multicriteria method all the variables involved are considered, thus, determining their true influence on the process.

Although the processes of development and social learning are not usually linked to the application of mathematics, this research shows how this science, applied through mathematical modeling [62], can also contribute in this regard by indicating or highlighting aspects that must be taken into account when evaluating and prioritizing projects, which were added or inserted in a multi-criteria decision-making process like the proposed ANP method. It can be of great help for decision makers and results in better benefits and a better quality of life for the population [25].

The above reasons mean that the integration achieved in the proposed methodology results in an innovative alternative that adapts to scenarios where the problems are usually complex as they involve society. Even more, considering that these communities are in transition from the conflict to the post-conflict period, the problems continue to exist and continue to require the attention of local and national governments and other institutions that work in the region. It is then observed how mathematics adapt to problems of a complex and real nature that not only allow to produce theory, also it is possible to put them into practice for the solution or the contribution to social development, especially in the communities within a post-conflict period.

## 8. Conclusions

According to the main goal raised from the beginning, this research allowed us to demonstrate that the use of participatory, numerical, and analytical tools integrated into a methodology for evaluating and prioritizing green projects facilitates decision-making regarding warning of the degree of affectation that investment decisions have on the natural resources of water and ground in addition to the acceptance or rejection that these projects will have thanks to citizen factors.

However, we can see that two different methods such as system dynamics and the analytical network process can be conveniently integrated and complemented with the purpose of making more appropriate decision-making regarding the selection of green projects for a community. In this case, modeling and simulation tools that DS offers are used like data provided to experts who were able to make informed decisions according to the peer review required within the ANP method. In this way, they are linked in the proposed methodology, citizen participation (participatory workshops and citizen factors), mathematical methods, and models for the representation of a part of the real world (system dynamics) and the multicriteria method for decision making.

Moreover, the early warnings generated by the model demonstrate the remarkable relationship between social development and the scarcity of natural resources. In this sense, we can see how the inappropriate use of resources also has adverse effects on the community, slows down growth processes, and deteriorates the quality of life of its inhabitants.

Furthermore, the results show the usefulness of applied mathematics in decision-making processes and in matters of social development where, therefore, environmental problems related to the appropriate use of natural resources are involved. In this case in particular, the applied methodology activated the processes of the social learning of the community of the California county and allowed the best choice to be made for the community between four different projects.

We also concluded that the component called "Citizen Factor" makes the problems and needs of the population visible and, at the same time, improves decision making since this factor establishes direct relationships between local knowledge (social, environmental, and economic) and the potential benefits of the projects that undergo the evaluation and prioritization process.

In the same way, it is evidenced in variables such as the reserve margins (of water and soil), available budget, and the incentives for green projects, the direct relationship between the social development of the community, as confirmed in the different ANP evaluations made by the experts where they are indicated as those with the most weight when prioritizing the projects.

On the other hand, it is demonstrated how the proposed methodology contributed to the reduction in biases in decision making in two moments, the first during participatory modeling involving different social actors and the second in the evaluation by pairs of the experts carried out in the ANP method.

In the same way, the weighting of the ANP method results allows to conclude that the citizen participation in the prioritization and evaluation processes goes beyond the needs of the moment and shows not only what is urgent and requires immediate action but also what is important and what is expected to provide a solution in the future. The differentiation of the urgent and the important allowed the weighting of the ANP results in the final evaluation.

Nevertheless, some limiting factors can affect the execution of the methodology, among them obtaining the active participation and trust of the community so the information obtained could be trustable and true. Likewise, the understanding of the scenarios presented to the experts and their relationship with the projects to be evaluated and prioritized constitutes an aspect to be treated with care, so that information is not lost or misinterpreted which could alter decision-making.

Likewise, the inequality that is currently perceived in the countryside prevents development processes from taking place properly. In the same way, the lack of support

and continuity of the projects by the governmental institutions, just as corruption causes the gap between the rural and the urban to be larger every day and that opportunities to improve the quality of life of the inhabitants, jobs, health, and education are lost, the foregoing given, among other things, by the stigma that has been woven with respect to the rural field and the individuals experiencing the conflict.

Contrary to that, in the California county such circumstances have been an engine that drives daily the search for change and improvement. The community is aware of its role in the territory and should not be alone in the search for well-being, especially if one takes into account that state support would consolidate the processes that are carried out.

Finally, this type of process, like the one followed in this research, generates discussions about the concept of development and the need to understand that development is not linear, nor can it be generalized. On the contrary, it is a process that is built with the transfer of the communities, their interests, conceptions, needs, views of the world, and the natural, economic, and cultural context. Therefore, it is necessary to stop seeing communities as homogeneous societies and to support them so that from their own knowledge they build their own forms of development that allow the achievement of the integral well-being of their inhabitants and of the territory.

**Author Contributions:** Conceptualization, J.A.C.-G. and G.O.-T.; data curation, G.O.-T. and J.V.-C.; formal analysis, J.A.C.-G. and J.V.-C.; funding acquisition, J.V.-C.; investigation, J.A.C.-G.; methodology, J.A.C.-G.; resources, G.O.-T.; validation, J.V.-C.; visualization, G.O.-T. and J.V.-C.; writing—original draft, J.A.C.-G.; writing—review and editing, J.A.C.-G. All authors have read and agreed to the published version of the manuscript.

**Funding:** This research was funded by Ministerio de Ciencia Tecnología e Innovacion de Colombia through National Doctorates program (announcement n°727 of 2015).

**Institutional Review Board Statement:** Not applicable.

**Informed Consent Statement:** Not applicable.

**Data Availability Statement:** Not applicable.

**Acknowledgments:** The authors are especially grateful to the National Doctorates program (announcement n°727 of 2015) of COLCIENCIAS, now the Ministry of Science, Technology and Innovation of Colombia. On the other hand, the authors express thanks to the National University of Colombia, likewise the Aysén University of Chile for their support for this type of research. Julián Castrillón Gómez expresses thanks to the California county for their determined collaboration and participation during the development of the workshops.

**Conflicts of Interest:** The authors declare no conflict of interest.

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
