# Peer review of "Systems Dynamics and the Analytical Network Process for the Evaluation and Prioritization of Green Projects: Proposal That Involves Participative Integration"

_sustainability, doi:10.3390/su141811519_

Round 1
Reviewer 1 Report (New Reviewer)
A well preserved through provoking review on evaluating green project in post war development in Colombia. It is time sensitive and most needed at this hour. Few specific comments are mostly technical and very little concerns:
In Abstract:
I have a problem with the statement “curious data of the state” which can be modified and/or rephrase.
In the statement. About the area description. Better change it to: Like California county in the Maglenda region of Colombia.
Overall technical: The whole manuscript suffers two different font color. Gray and pure black. Fix it.
Figures: The figures must be updated with larger fonts for the description with higher resolution, some of them even unreadable (we cannot expand it even).
Authors Contribution: Do all equally contributed or have them in parts. Needs to be mentioned.
Figure legends should be more elaborate and speaks about the figure itself. Also The table below the Figure 20 to be reformatted.
Any references beyond 2020s?
Author Response
We wish to thank the anonymous reviewer for her/his careful reading of the manuscript that led to a substantial revision of the paper
Comments Reviewer 1:
A well preserved through provoking review on evaluating green project in post war development in Colombia. It is time sensitive and most needed at this hour. Few specific comments are mostly technical and very little concerns:
- In Abstract:
I have a problem with the statement “curious data of the state” which can be modified and/or rephrase.
Reply:
The reviewer is right and we want to thank for the point mention. It corresponds to a writing error in the text in the English language. Now, we have modified the statement “curious data of the state” by “particular data of the zone”.
- In the statement:
About the area description. Better change it to: Like California county in the Maglenda region of Colombia.
Reply:
We want to thank the reviewer for your kind suggestion. Now, we have modified the Figure 18 in the manuscript so that the county name actually appears as California county.
- Overall technical: The whole manuscript suffers two different font color. Gray and pure black. Fix it.
Reply:
We want to thank the reviewer for your kind suggestion. Now, we will ensure that the final version of the manuscript appears with the same Font color.
- Figures: The figures must be updated with larger fonts for the description with higher resolution, some of them even unreadable (we cannot expand it even).
Reply:
We thank the anonymous reviewer for pointing out this important aspect. Now, we have updated the figures with larger fonts for the description, and we will ensure that the final version of the manuscript the figures appears with higher resolution possible.
- Authors Contribution: Do all equally contributed or have them in parts. Needs to be mentioned.
Reply:
We thank the anonymous reviewer for pointing out this important aspect. Now, we have included in the manuscript the authors contributions as follow:
Conceptualization, Julian Andres Castrillon-Gomez and Gerard Olivar-Tost; Data curation, Gerard Olivar-Tost and Johnny Valencia-Calvo; Formal analysis, Julian Andres Castrillon-Gomez and Johnny Valencia-Calvo; Funding acquisition, Johnny Valencia-Calvo; Investigation, Julian Andres Castrillon-Gomez; Methodology, Julian Andres Castrillon-Gomez; Resources, Gerard Olivar-Tost; Validation, Johnny Valencia-Calvo; Visualization, Gerard Olivar-Tost and Johnny Valencia-Calvo; Writing – original draft, Julian Andres Castrillon-Gomez; Writing – review & editing, Julian Andres Castrillon-Gomez.
- Figure legends should be more elaborate and speaks about the figure itself. Also The table below the Figure 20 to be reformatted.
Reply:
We appreciate the observation of reviewer 1. Now, we have reviewed all the figures in the manuscript to rename those that need more elaborate legends and that allow the figures speak by themselves. Additionally, the Table 1 have been reformatted in order to fullfilment the journal standards.
- Any references beyond 2020s?
Reply:
We thank the anonymous reviewer for pointing out this important aspect. Now, it is important to clarify that this article is the product of continued work that has been proposed in research previously published in this and other journals. Likewise, the previous works found in the literature are duly referenced in order to adequately justify this manuscript. The previous own works to which we refer are:
[6] G. Olivar-Tost, J. Valencia-Calvo, and J. A. Castrillón-Gómez, “Towards decision-making for the assessment and prioritization of green projects: An integration between system dynamics and participatory modeling,” Sustain., vol. 12, no. 24, pp. 1–23, 2020.
[7] J. A. Castrillón Gómez and J. Valencia Calvo, “Propuesta de modelo en dinámica de sistemas para toma de decisiones en selección de proyectos verdes,” Rev. Mutis, vol. 8, no. 2, pp. 84–94, 2018.
In relation to the literature after 2020, as mentioned in the manuscript, not many investigations have been found in this regard where the aforementioned methodologies are integrated and it has not been possible to find any in which such integration is used for the evaluation and prioritization of projects in the way proposed in this article.
Reviewer 2 Report (New Reviewer)
The manuscript seems to be interesting in the field. I found most of the figures are not visible in pdf form of the manuscript. should be arranged in better. However, can be accepted after minor revision.
Author Response
We wish to thank the anonymous reviewer for her/his careful reading of the manuscript that led to a substantial revision of the paper.
Comments Reviewer 2:
The manuscript seems to be interesting in the field. I found most of the figures are not visible in pdf form of the manuscript. should be arranged in better. However, can be accepted after minor revision.
Reply:
We appreciate the interest of reviewer 2 in the manuscript. Now, we have updated and organized the figures with larger fonts for the description, and we will ensure that the final version of the manuscript the figures appears with higher resolution possible.
Reviewer 3 Report (New Reviewer)
This paper proposes a methodology for evaluating and prioritizing green projects based on an integrated approach between System Dynamics Modeling and the Analytical Network Process. The study is interesting and valuable conclusions are drawn. The manuscript has a certain dose of intrinsic merit but is not publishable as it stands. I have several comments which may help the authors to further improve the quality for publication standard.
Comments and Suggestions:
1. The introduction, though industrious, fails to highlight one very relevant aspect, namely the original contributions of the manuscript. The authors hint at these in other sections but should offer a clear discussion in this area.
2. This paper needs to supplement the related literature review of the application of system dynamics.
3. What's the Green Project? What is the definition and scope of it?
4. The picture need to be optimized. Take Figure 4 as an example. In the legend, "Current" and "95%" are both blue. It is difficult for the reader to distinguish the specific scope of the two in the picture. It is suggested to replace "current" with another colors (such as red) different from that in the picture.
5. The information in Table 1 is disorganized, and the authors may need to adjust the indicators according to different dimensions, and list the related survey results.
6. In this paper, the expert scoring method is used to determine the priority order of project indicators, but the author does not disclose any expert information, which will arouse readers' doubts about the evaluation results. It is suggested that the authors show the relevant expert information and grading details, such as the number of expert, age distribution, occupation or research direction, unit, education background, etc., which can be displayed in the attached table.
7. Any limit? In this paper, system dynamics and network analysis process are used to solve the ranking problem of the two elements (water and land), but is this method still applicable if other indicators (such as energy and food) are added? That is, can the model solve the ranking problem of multiple factors (water, land, energy and food)?
Author Response
We wish to thank the anonymous reviewer for her/his careful reading of the manuscript that led to a substantial revision of the paper.
Comments Reviewer 3:
This paper proposes a methodology for evaluating and prioritizing green projects based on an integrated approach between System Dynamics Modeling and the Analytical Network Process. The study is interesting and valuable conclusions are drawn. The manuscript has a certain dose of intrinsic merit but is not publishable as it stands. I have several comments which may help the authors to further improve the quality for publication standard.
Comments and Suggestions:
- The introduction, though industrious, fails to highlight one very relevant aspect, namely the original contributions of the manuscript. The authors hint at these in other sections but should offer a clear discussion in this area.
Reply:
We thank reviewer 3 for his concern in this important aspect. Now, it is important to clarify that in the introduction, emphasis is placed on the proposed methodology and, on the interrelationship analysis between the references and how these ideas and strategies contributed to the implementation of the model and, in general, of the entire methodology. On the other hand, we understand the reviewer's concern in relation to the relevant aspects and the original contributions of the manuscript, therefore, in section 7, an extensive discussion is presented where the related literature is contrasted with the proposed methodology and, in addition, the most important aspects of the research are highlighted. Finally, in the conclusions, the findings and contributions made to the literature are presented, as well as the work that could be developed in the future.
- This paper needs to supplement the related literature review of the application of system dynamics.
Reply:
We thank the anonymous reviewer for pointing out this important aspect. Now, it is important to clarify that this article is the product of continued work that has been proposed in research previously published in this and other journals. Likewise, the previous works found in the literature are duly referenced in order to adequately justify this manuscript. The previous own works to which we refer are:
[6] G. Olivar-Tost, J. Valencia-Calvo, and J. A. Castrillón-Gómez, “Towards decision-making for the assessment and prioritization of green projects: An integration between system dynamics and participatory modeling,” Sustain., vol. 12, no. 24, pp. 1–23, 2020.
[7] J. A. Castrillón Gómez and J. Valencia Calvo, “Propuesta de modelo en dinámica de sistemas para toma de decisiones en selección de proyectos verdes,” Rev. Mutis, vol. 8, no. 2, pp. 84–94, 2018.
And some references directly related to the research topic found in the literature are:
[14] M. Bottero, G. Datola, and E. De Angelis, “A System Dynamics Model and Analytic Network Process : An Integrated Approach to Investigate Urban Resilience,” pp. 24–26, 2020.
[49] X. Xi and K. L. Poh, “A Novel Integrated Decision Support Tool for Sustainable Water Resources Management in Singapore: Synergies Between System Dynamics and Analytic Hierarchy Process,” Water Resour. Manag., vol. 29, no. 4, pp. 1329–1350, 2014.
[55] K. Yu, Q. Cao, C. Xie, N. Qu, and L. Zhou, “Analysis of intervention strategies for coal miners ’ unsafe behaviors based on analytic network process and system dynamics,” Saf. Sci., vol. 118, no. June 2018, pp. 145–157, 2019.
[51] S. Elsawah, D. Danesh, and M. Ryan, “A strategic asset planning decision analysis : An integrated System Dynamics and multi-criteria decision-making method,” no. Iso 55000, 2019.
[26] R. Sayyadi and A. Awasthi, “An integrated approach based on system dynamics and ANP for evaluating sustainable transportation policies,” Int. J. Syst. Sci. Oper. &Logistics, vol. 0, no. 0, pp. 1–10, 2018.
Similarly, as mentioned in the document, not many investigations have been found in this regard where the aforementioned methodologies are integrated and it has not been possible to find any in which such integration is used for the evaluation and prioritization of projects in the way proposed in this article.
- What's the Green Project? What is the definition and scope of it?
Reply:
We thank the anonymous reviewer for pointing out this important aspect. Now, when talking about green projects, it refers to sustainable initiatives, where not only environmental protection stands out, but also efforts are coordinated to improve the social, economic and well-being conditions for these post-conflict communities and, always under the principles of participation that lead to sustainable development. In this regard, we have modified the second paragraph in the introduction of the manuscript to clarify, not only sustainability but also green projects.
The paragraph:
However, as we can see in [3], when we talk about sustainability the importance is highlighted, not only for protecting the areas of environmental interest, also for articulat-ing efforts to improve the labor, economic and welfare conditions for men and women be-longing to these communities so we can contribute to reduce the gap between urban and rural áreas.
Its amended as follow:
However, as we can see in [3], when we talk about sustainability and green projects, the importance is highlighted, not only for protecting the areas of environmental interest, also for articulat-ing efforts to improve the labor, economic and welfare conditions for men and women be-longing to these communities so we can contribute to reduce the gap between urban and rural áreas.
- The picture need to be optimized. Take Figure 4 as an example. In the legend, "Current" and "95%" are both blue. It is difficult for the reader to distinguish the specific scope of the two in the picture. It is suggested to replace "current" with another colors (such as red) different from that in the picture.
Reply:
We thank the anonymous reviewer for pointing out this important aspect. Now, we have updated the figures with larger fonts for the description, and we will ensure that the final version of the manuscript the figures appears with higher resolution possible.
- The information in Table 1 is disorganized, and the authors may need to adjust the indicators according to different dimensions, and list the related survey results.
Reply:
We appreciate the observation of reviewer 1. Now, Table 1 has been revised and reformatted in order to better appreciate the findings identified through the variety of tools used with the community and their classification in the different proposed dimensions.
- In this paper, the expert scoring method is used to determine the priority order of project indicators, but the author does not disclose any expert information, which will arouse readers' doubts about the evaluation results. It is suggested that the authors show the relevant expert information and grading details, such as the number of expert, age distribution, occupation or research direction, unit, education background, etc., which can be displayed in the attached table.
Reply:
We thank reviewer 3 for her/his important appreciation. Now, it is important to clarify two aspects, first, the objective of the proposed methodology is to provide a tool or platform in which a person can review relevant information about the community, such as their interests, problems, availability of resources, among others, and thus prioritize a decision. Second, the concept of expert (decision maker) may vary from one region to another, even from one country to another, depending on the context of application. In a centralized country like Colombia, for example, decisions are made in the central government, with professionals who are part of these units but who are unaware of the reality of the territories where the projects are applied. Therefore, in the manuscript, the methodological development and its role as a decision-making tool are shown, and how this tool can be addressed to experts from central organizations or research agencies so that it can be used to facilitate the decision-making process about the evaluation and prioritization of green projects.
- Any limit? In this paper, system dynamics and network analysis process are used to solve the ranking problem of the two elements (water and land), but is this method still applicable if other indicators (such as energy and food) are added? That is, can the model solve the ranking problem of multiple factors (water, land, energy and food)?
Reply:
We want to thank reviewer 3 for her/his important question. Indeed, thanks to the integration proposed in this methodology, the model can be extrapolated to other processes such as those mentioned (energy, food, among others), as long as the dynamic hypotheses of each of these processes are available, so that they can be integrated into the system.
This manuscript is a resubmission of an earlier submission. The following is a list of the peer review reports and author responses from that submission.
Round 1
Reviewer 1 Report
This paper proposes a methodology of evaluation and prioritization of projects based on an approach integrated between the modelling in dynamics systems (SDM) and the Analytical process network (ANP), in which the citizen factors are used as qualitative and quantitative variables inside the posed balance sheet model. After going through the paper, I found some concerns, as listed below:
1. I suggest reducing the title a bit.
2. An abstract is normally between 200 to 250 words, thus a concise and factual abstract is required to state briefly the purpose of the research, the principal results and the major conclusions. Add some of the most important quantitative results to the abstract. Focus on the advantages of the proposed method with respect to the obtained results.
3. Most of the ideas written were already described in many literatures. The Authors tried to compile it but lack of the enhancement of the interrelation analysis between the references. It is advised that the authors give a deeper analysis on how these ideas become more applicative strategies so that they can contribute to the next step of implementation.
4. Avoid lumping references such as [23-26]. Instead summarize the main contribution of each referenced paper in a separate sentence and by including the reference number.
5. The novelty of the present work should be well stated and justified. The new author's contribution should be justified regarding the previous works in the literature. For example, https://doi.org/10.3390/pr9081375.
6. More in-depth analysis of the author's contribution of this paper in the introduction section. Moreover, I would like to see more discussion of the literature so that I can clearly identify the article relates to competing ideas.
7. The conclusion is a bit general or it lacks distinction. Please refine this further to be more thought-provoking. Moreover, I don't recommend adding references in the conclusion.
8. Too many citations for T. L. Saaty, I suggest to remove some.
Reviewer 2 Report
I think that this manuscript is not suitable for publication in the selested journal. Maybe in some other, after extensive remodeling.
General notes:
- English is good.
- The figures are not visible in their entirety. The writing in the pictures is tiny and unintelligible. The scenes in Figures are incomprehensible.
- The article is not consistent.
- The solved problems depends on region. The model is not transferable (as the authors themselves point out).
Reviewer 3 Report
I am very sorry but figures 1 to 16 are not displayed correctly in the manuscript so it is difficult to evaluate it